
# Temperature profiles, plumes and spectra in the surface layers of convective atmospheric boundary layers

Keith G. McNaughton[1] and Subharthi Chowdhuri[2]

[1]Independent Researcher, Kerikeri, New Zealand
[2]Indian Institute of Tropical Meteorology, Pune, Maharashtra 411008, India

**Correspondence:** K.G. McNaughton (keith@mcnaughty.com)

**Abstract.** We survey temperature patterns and heat transport in convective boundary layers (CBLs) from the perspective that these are emergent properties of far-from-equilibrium, complex dynamical systems. We use the term 'plumes' to denote the temperature patterns, in much the same way that the term 'eddies' is used to describe patterns of motion in turbulent flows. We introduce a two-temperature (2T) toy model to connect the scaling properties of temperature gradients, temperature variance and heat transport to the geometric properties of plumes. We then examine temperature ($T$) probability density functions and $w$-$T$ joint probability density functions, $T$ spectra and $wT$ cospectra observed both within and above the surface friction layer. Here $w$ is vertical velocity. We interpret these in terms of the properties of the plumes that give rise to them. We focus first on the self-similarity property of the plumes above the SFL, and then introduce new scaling results from within the SFL, which show that $T$ spectra and $wT$ cospectra are not self-similar with height at small heights $z/z_s < 0.1$, but increasingly display properties associated with random diffusion. The CBL similarity parameters defined by McNaughton et al (Non-linear Processes in Geophysics 14, 257-271, 2007) are used throughout. We conclude by contrasting our interpretation of the role of buoyancy in CBL flows with that of Richardson (proc. Roy. Soc. London A 87, 354-373, 1920), whose ideas inform the current interpretation of the statistical fluid mechanics model of boundary-layer flows.

## 1 Introduction

Turbulent transport in convective atmospheric boundary layers depends on the forms, sizes and energies of the patterns of motion—'coherent structures' and 'eddies'—found within them. These patterns can be regarded as emergent properties of convective boundary layer (CBL) flows. When heat is introduced at the upper and lower boundaries of such flows these eddies create patterns of temperature, or 'plumes', that are also emergent properties of these flows. Our task is to describe these eddies and plumes, and their inter-relationships. The problem is a difficult one, and must necessarily be based on experiment since the governing equations cannot be solved. Though experimental observations and results from simulations have been analyzed in many ways, and much has been learned, the most systematic way to begin is by scaling analysis. That is, we must learn





to present these observations in universal ways. Success in finding appropriate scaling parameters will go hand-in-hand with understanding what controls the sizes of, energy and scalar concentrations in, and fluxes carried by particular classes of eddies

and plumes. The forms of the eddies and plumes must be discovered by other means.

In previous work we have focussed on the scaling properties of spectra and cospectra, particularly of temperature ($T$) spectra and $wT$ cospectra from convective boundary layers (McNaughton et al, 2007; Laubach and McNaughton, 2009; Chowdhuri et al, 2019), where $w$ is vertical velocity. The reason is that spectra naturally sort eddies by size, since the essential non-linearity of the governing Navier-Stokes equations denies linear superposition of eddies, so eddies of different kinds cannot coexist

unless they are well-separated by size. Implicit in this argument is that only a small number of distinct emergent patterns of motion can satisfy all the feedback requirements inherent in this flow system. Empirical results support this proposition (Perry and Abell, 1975; Perry et al, 1986; McNaughton et al, 2007).

The rationale for our basic set of scaling parameters is given in earlier papers (McNaughton, 2004; McNaughton et al, 2007; Laubach and McNaughton, 2009). The set has two fundamental length parameters: $\lambda$, representing the size of the largest

turbulent structures in the CBL, and $z$, representing observation height. It has two parameters to describe the flow of mechanical energy through the CBL system: the outer dissipation rate, $\epsilon_o$, which is constant with height above the surface friction layer (SFL) (Kaimal et al., 1976; McNaughton et al, 2007), and the dissipation velocity, $u_\epsilon$. The latter, with $z$, parameterizes the dissipation rate within the SFL. In practical terms, $\lambda$ is the peak wavelength of the streamwise velocity ($u$) spectrum, and $u_\epsilon = (kz\epsilon)^{1/3}$, where $\epsilon$ is the dissipation rate at height $z$ near the ground, and $k$ is the von Kármán constant. We also use the

height of the SFL, $z_s$, given by

$$z_s = \frac{u_\epsilon^3}{k\epsilon_o} \tag{1}$$

The SFL is characterized by the presence of a special class of attached shear eddies. These develop upwards from the ground until their growth is terminated by interaction with detached eddies of similar size from the outer Kolmogorov cascade. Equation (1) represents the height at which inner and outer dissipation rates match, as explained by McNaughton (2004) and

McNaughton et al (2007). Finally, we use the kinematic heat flux $H$, to construct a temperature scale. Buoyancy effects of the heat flux are accounted through the dissipation rates.

What complicates the picture is that the basic length scales, $\lambda$, $z$ and $z_s$, appear not just alone but also in combinations, as mixed length scales and doubly-mixed length scales. A mixed scale is the geometric mean of two component length scales, so taking the form $\ell_1^{1/2}\ell_2^{1/2}$ where $\ell_1$ and $\ell_2$ are the component length scales. Doubly-mixed scales follow the same pattern, but

with one of the component scales itself being a mixed scale. We observe that mixed scales are found when one kind of eddy or plume, with length scale $\ell_1$ exists within, and has sizes or aggregation properties organized by larger eddies with length scale $\ell_2$. It has been found that mixed and doubly-mixed scales often describe turbulence processes near smooth walls (e.g., Alfredsson and Johansson, 1984; DeGraaff and Eaton, 2000; Metzger et al., 2001; Buschmann et al, 2009), with half powers appearing in every case, even while dimensional consistency requires only that a scale length takes the form $\ell_1^\alpha\ell_2^{1-\alpha}$, where

$\alpha$ has any value. The possibility of mixed scales puts the search for suitable scales beyond the reach of simple dimensional analysis.





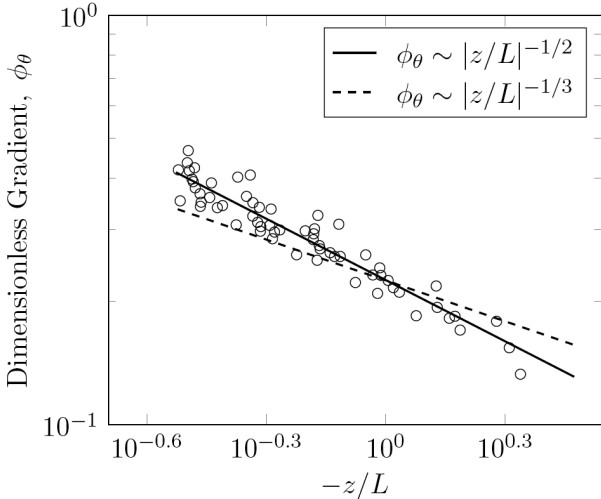

**Figure 1.** Dimensionless potential temperature gradient plotted against height, with each axis scaled according to the Monin-Obukhov similarity model so that $\phi_\theta = (z u_*/H)\partial\theta/\partial z$, and $z$ is scaled on the Obukhov length $-L$. The $-1/2$ power law describes the data over two decades of $-z/L$. Redrawn from Businger et al (1971).

In this paper we survey the scales found to be successful in collapsing temperature ($T$) spectra and $w$-$T$ cospectra from above and within the SFL, and interpret them in terms of the known properties of eddies and plumes found in CBLs. We include a discussion the temperature profile in this survey, because its gradient follows a $-1/2$ power law for more than a decade, from mast top down to deep within the SFL, as shown in Fig. 1. Power laws are of interest in any complex system because they indicate a scale invariance and self-similarity of the underlying structures, and we find half power laws to be particularly interesting now that we have become familiar with mixed scales. Generally, we seek to explain the ensemble-averaged descriptions of these flows, as observed in experiments, in terms of the known properties of the underlying, emergent structures that exist in space and time in CBL flows. That is, we seek to extend our conceptual model of CBL turbulence and its transport properties beyond what we have attempted previously.

## 2 Temperature gradients

We begin with mean temperature gradients in the atmospheric surface layer of CBLs since they are important in themselves, and they open an important window onto a self-similarity property of plumes near the ground. The temperature gradient in much of the surface layer of CBLs can be written as

$$\frac{z u_*}{H}\frac{\partial\overline{\theta}}{\partial z} = -\alpha\left(\frac{z}{-L}\right)^{-1/2} \tag{2}$$





using the scaling parameters proposed by the Monin-Obukhov similarity theory (MOST) (Monin and Obukhov, 1954). We use the term 'surface layer' to mean the layer below the mixed layer in CBLs, or roughly the bottom 10% of the CBL. Our usage is compatible with MOST usage, where the surface layer is defined as the layer where heat and momentum fluxes are sufficiently constant with height that these fluxes can be represented by their surface values within the surface layer. The SFL is a sub-layer

of the surface layer in CBLs. In (2) $u_*$ is the friction velocity, $H$ is the kinematic heat flux at the ground, $\overline{\theta}$ is time-averaged potential temperature, $\alpha$ is a constant and $L$ is the Obukhov length, defined by

$$-L = \frac{u_*^3 T_0}{kgH} \tag{3}$$

where $g$ is the acceleration due to gravity and $T_0$ is average temperature in degrees K. The $-1/2$ power in (2) was established empirically and is not consistent with MOST itself, which predicts that the profile should approach a $-1/3$ power law in the

limit of windless convection, as $-z/L \to \infty$ (Businger et al, 1971). The comparison is shown in Fig. 1. We note that Kader and Yaglom (1990), henceforth KY90, fit two segments of $-1/3$ power laws to their data from several sources, in conformity with their three-layer model of the surface layer and the requirements of MOST, even though the $-1/2$ power law provides a better fit over the full range $0.04 < -z/L < 4$ (KY90: Fig.1). Foken and Skeib (1983) report that the data from the Tsimlyansk site in 1981 follow a $-1/2$ power law for all $-z/L > 0.06$.

The scaling used in (2) is not unique. Power laws indicate scale-free, self-similar behavior of the underlying structures giving rise to the power law, so temperature profiles can be scaled in whatever way best suits the context. Thus the scales in (2) can be replaced by those that Laubach and McNaughton (2009) found to give universal forms to $T$ spectra and $wT$ cospectra within the SFL. Using these scales we have

$$\frac{z(z_s\epsilon_o)^{1/3}}{H} \frac{\partial\overline{\theta}}{\partial z} = -\alpha_1 \left(\frac{z}{z_s}\right)^{-1/2} \tag{4}$$

where $\alpha_1$ is another positive constant. Here the value of $u_\epsilon$, used to calculate $z_s$ by (1), is calculated from $u_\epsilon = (kz\epsilon)^{1/3}$ where the dissipation rate $\epsilon$ is measured deep within the SFL, so $u_\epsilon$ is like $u_*$ but includes the effect of very large eddies on dissipation rates near the ground, as first discussed by Townsend (1961). Unlike $L$, $z_s$ does not go to zero in windless convection.

We now have two equations, both describing the same temperature profile but scaled in two different ways. Both can achieve some success if the parameters of the two models are statistically correlated. Results from SLTEST (Chowdhuri et al, 2019) do

show a strong correlation between $z_s$ and $L$. If this is generally true then both (2) and (4) can be expected to give fairly good accounts of observed temperature profiles, despite their very different foundations. The question is which scaling scheme gives less scatter and the more 'universal' results?

We await direct experimental comparisons, but in the meanwhile we find it more convenient to work with (4) because its parameters are related to the geometrical properties of particular classes of eddies and plumes in CBLs, and it is these properties

that we investigate here. Before starting we rewrite (4) as

$$\frac{\partial\overline{\theta}}{\partial\zeta} = -\alpha_1\theta_*\zeta^{-3/2} \tag{5}$$

where $\zeta = z/z_s$ is dimensionless length, $\theta_*$ the temperature scale $H/(z_s\epsilon_o)^{1/3}$ and $\alpha_1$ a constant.





An interesting question is the plausibility of the profile forms (2) and (4) above the SFL. KY90 discuss a three-layer model of the lower CBL, and they find it surprising that (2) applies in their convective sublayer where $u_*$ is not a relevant parameter,

reasoning that since $L$ depends on $u_*$ then $L$ should have no significance there. Equally, our profile expressions, (4) and (5), employ the length scale $z_s$, which we would not expect to be relevant above the SFL. In our case the problem is resolved when we note that the height variable in (5) can be rescaled by replacing $z_s$ with the outer length scale, $\lambda$. We can then redefine the scaled height variable as $\zeta = z/\lambda$, and redefine the temperature scale as $\theta_* = H/(\lambda\epsilon_o)^{1/3}$ in (5). With these we can rewrite (5) in a form identical to (5) but with $\alpha_1$ redefined as $\alpha_2 = \alpha_1(z_s/\lambda)^{1/6}$. In this new version of (5) the length scale $\lambda$ reflects

the streamwise extent of the largest eddies found in CBLs, so it is closely related to the inversion height, $z_i$. The ratio of $z_i$ to $\lambda$ will depend on the aspect ratio of the large cellular or roll structures in the CBL, and this relationship has not been fully explored. We note that Kaimal et al. (1976) reported that $\lambda = 1.5z_i$ for thirteen runs from the five days before a lightening strike terminated the Minnesota experiment. Overall, we see that (5) can be written in either inner-scaled or outer-scaled versions.

KY90 also present profiles of the standard deviation of temperature, $\sigma_\theta$, observed during many experiments, both within

and above their 'convective-dynamic sublayer'. The top of this sublayer corresponds well to the top of our SFL, but our SFL extends right down to the ground. We will return to this later, but here we note that KY90 plot their profiles within and above the SFL in two segments using MOST scaling parameters, and find that

$$\frac{u_*\sigma_\theta}{H} = \beta\left(\frac{z}{-L}\right)^{-1/3} \tag{6}$$

in each, though the constant $\beta$ has a different value in each segment. The $-1/3$ power is consistent with expectation from

MOST in the limit of free convection and with the earlier results of Priestley (1960); Businger et al (1971) and Wyngaard and Coté (1972). This behavior extends down to $-z/L \approx 0.05$. The side-step in the profile of KY90 is unexplained.

Once again, this profile can be written in terms of our favored parameters, giving

$$\sigma_\theta = \beta_1\theta_*\zeta^{-1/3} \tag{7}$$

where $\zeta = z/\lambda$. This profile agrees with the variances found by integrating the areas under scaled temperature spectra observed

above the SFL (McNaughton et al, 2007; Chowdhuri et al, 2019), and also within it according to the new $T$ spectra presented below, which self-similarity extends down to similarly low levels.

To summarize: temperature profiles follow a $z^{-1/2}$ power law through most of the surface layer, while profiles of its standard deviation, $\sigma_T^2$, follows a $-1/3$ power law over a similar range. Since both profiles follow power-laws we infer that these profiles are associated with some kind of scale-free, self-similar structures. In the lowest part of the SFL the profiles depart from these

power law forms and the temperature profile follows a log law ($\partial\theta/\partial z \propto z^{-1}$) while the $\sigma_\theta$ profile becomes constant with height, which behavior indicates that heat transport there is effected by another kind of self-similar eddy structure.

In searching for the mechanisms that underly these mean profiles we will appeal to the geometric and scaling properties of various kinds of plumes, and to do this we must have a working understanding of the nature of plumes.



## 3 Modeling plumes

We use the word 'plume' to mean any connected volume of air defined by its temperature, so our usage goes beyond the narrow regions of ascending air that common usage implies. Plumes may frequently take this form, but we include all warm and cool updrafts and down-drafts, independent of their shape or size. We restrict our discussion to heights above any viscous sublayer or roughness sublayer immediately above the ground, and to below the mixed layer. That is, we deal with the atmospheric surface layer as it is usually defined in boundary-layer meteorology.

Plumes, like eddies, defy exact definition. They are recognizable 'patterns of scalar concentration', just as eddies are 'patterns of motion', and they can have discoverable scaling properties even while the patterns themselves are more-or-less ill-defined. There is an intuitive connection between a plume or an eddy and the concept of a strange attractor, since the actual shapes of plumes and eddies will exhibit many variations on a theme. Since we find the concept of plumes, and the shapes of plumes to be useful concepts, we now introduce a toy model in which plume boundaries, and so plume shapes, can be rigorously defined.

### 3.1 A two-temperature plume model


Our 2T model is a toy model that allows temperatures to have just one of two values at each point while allowing the velocity field to have all of its observed properties. This is a great simplification of real temperature fields, and it does create some problems, but this 2T model also leads to some simple relationships that will help us to understand the similarity properties of real plumes observed in real boundary layers.

We consider a plane horizontal flow in which there are two kinds of air, each distinguished by its temperature, $\theta_u$ or $\theta_d$. Each kind is organized into localized regions called plumes. Warmer plumes originate somewhere below and move upwards on average, while cooler plumes originate somewhere above and generally move downwards. We neglect molecular diffusion, so the up-and down-plumes retain the distinct identities imparted to them at their sources, even while they may be stretched and folded in complicated ways. Warm plumes move upwards with an mean velocity $\langle w_u \rangle$, and cool plumes move downwards

at $\langle w_d \rangle$, where the angle brackets indicate means over the areas of each kind of plume measured on a horizontal plane. A heat flux is maintained by air being converted from $\theta_d$ to $\theta_u$ in a thin layer adjacent to the ground, at heights below those of interest here. In the real world this heating would occur within a viscous sub-layer or roughness sub-layer adjacent to the ground, which layers are excluded from present consideration.

### 3.2 Relationships in the 2T model

The 2T model leads to a number of basic relationships.

### 3.2.1 Continuity

Let the warmer and cooler plumes occupy fractional areas $f_u$ and $f_d$ on any horizontal plane at height $z$. These fractions comprise the whole, so

$$f_u + f_d = 1 \tag{8}$$





There is no net vertical transport of air, so

$$\langle w_u \rangle f_u + \langle w_d \rangle f_d = 0 \tag{9}$$

### 3.2.2 Plume velocity variance

Because there are two mean vertical velocities there will be a velocity variance, and this will depend on the area fractions occupied by the up- and down-plumes at each height. Starting with

$$\sigma^2_{<w>} = \left\langle \langle w \rangle^2 \right\rangle - \langle \langle w \rangle \rangle^2 \tag{10}$$

we write

$$\begin{aligned}\sigma^2_{\langle w \rangle} =& f_u \langle w_u \rangle^2 + f_d \langle w_d \rangle^2 \\ & - (f_u \langle w_u \rangle + f_d \langle w_d \rangle)^2 \end{aligned} \tag{11}$$

giving

$$\sigma^2_{\langle w \rangle} = f_u f_d \left( \langle w_u \rangle - \langle w_d \rangle \right)^2 \tag{12}$$

which can also be written as

$$\sigma^2_{\langle w \rangle} = - \langle w_u \rangle \langle w_d \rangle \tag{13}$$

by using (9).

We note that (13) gives the variance of the plume mean velocities, not the total variance of vertical velocity, $\sigma^2_w$. This is an
important distinction, and we have no guarantee that $\sigma_w$ and $\sigma_{\langle w \rangle}$ will be equal, or even that they will scale the same way.

### 3.2.3 Temperature variance

Because the up- and down- plumes have different temperatures there is temperature variance on the $(x,y)$ plane. Starting with

$$\sigma^2_\theta = \left\langle \theta^2 \right\rangle - \langle \theta \rangle^2 \tag{14}$$

we write

$$\sigma^2_\theta = f_u \theta_u^2 + f_d \theta_d^2 - (f_u \theta_u + f_d \theta_d)^2 \tag{15}$$

and so

$$\sigma^2_\theta = f_u f_d \left( \theta_u - \theta_d \right)^2 \tag{16}$$

Thus $\sigma^2_\theta$ is proportional to $f_u f_d$. We note that $(\theta_u - \theta_d)$ is independent of height, so it may be regarded as the temperature scale in our 2T model. The product of the fractional areas, $f_u f_d$, will depend on height scaled by an appropriate height scale.





### 3.2.4 The heat flux

The kinematic heat flux is given by

$$\langle w'\theta' \rangle = \langle w_u \rangle f_u \theta_u + \langle w_d \rangle f_d \theta_d \tag{17}$$

where $\langle w'\theta' \rangle$ is averaged over the whole horizontal plane. We will use the symbol $H$ for this heat flux. Combining (17) with (9) gives

$$H = \langle w_u \rangle f_u (\theta_u - \theta_d) \tag{18}$$

and its twin

$$H = -\langle w_d \rangle f_d (\theta_u - \theta_d) \tag{19}$$

Since $H$ and $(\theta_u - \theta_d)$ are both constant with height we have an inverse relationship between $\langle w_u \rangle$ and $f_u$ and between $\langle w_d \rangle$ and $f_d$. On average the up-plumes get thinner and faster as they ascend, while the down-plumes get thinner and faster as they descend.

Multiplying left and right sides of the last two equations together we get

$$H^2 = -\langle w_u \rangle \langle w_d \rangle f_u f_d (\theta_u - \theta_d)^2 \tag{20}$$

which can be rewritten as

$$H^2 = \sigma_{\langle w \rangle}^2 \sigma_\theta^2 \tag{21}$$

using (13) and (16). The temperature variance scales on $H^2/\sigma_{\langle w \rangle}^2$.

### 3.2.5 The temperature gradient

The mean temperature on the horizontal plane at any value of $z$ is

$$\langle \theta \rangle = f_u \theta_u + f_d \theta_d \tag{22}$$

Using (8) we can write the mean temperature gradient as

$$\frac{\mathrm{d}\langle \theta \rangle}{\mathrm{d}z} = (\theta_u - \theta_d)\frac{\mathrm{d}f_u}{\mathrm{d}z} \tag{23}$$

or its twin

$$\frac{\mathrm{d}\langle \theta \rangle}{\mathrm{d}z} = -(\theta_u - \theta_d)\frac{\mathrm{d}f_d}{\mathrm{d}z} \tag{24}$$

We notice that the temperature difference $(\theta_u - \theta_d)$ is a constant, so the temperature gradient depends solely on how the fractional area of updrafts $\mathrm{d}f_u/\mathrm{d}z$, or its complement $\mathrm{d}f_d/\mathrm{d}z$, changes with height. If warm plumes become faster and thinner as they rise then area-mean temperature will decrease with height.





### 3.3 Limitations of the 2T model

The 2T model is a toy model, designed to help us to understand the link between plume geometry and mean vertical profiles of temperature and temperature variance. Its principal value is that it allows us to talk of the areas of plumes without worrying too much about how the boundaries of those plumes should be defined. However, the 2T approximation has its limitations.

We can explore these by considering the temperature signal along a horizontal transect, as represented by the 2T model, and comparing this with a more-realistic representation. A 2T transect would look like a comb with many teeth, all of the same height $(\theta_u - \theta_d)$ but with various widths and grouped into clusters of various sizes and compositions. A more realistic model would allow molecular diffusion to operate, rounding the corners of the teeth, thereby reducing the heights of smaller teeth with widths comparable to the Kolmogorov microscale length. Despite these changes the resulting groupings of fuzzy plumes would have almost the same mean temperature as their 2T originals. On the other hand the temperature variances would be reduced. For this reason we can have some confidence in (23) but we must be wary of (16) and relationships that follow from it.

Despite its limitations, the 2T model allows us to talk of the lengths and areas of plumes, and so to interpret the power-law temperature profile geometrically, in terms of the self-similar tapering with height of plume cross sections. We believe that this remains useful even while real plumes are blurry composites of whole populations of smaller plumes, and even while the plumes originating from near the ground and the top of the CBL have, in reality, a wide range of temperatures. Also, the 2T model explains how the profiles of $\partial\overline{\theta}/\partial z$ and $\partial\sigma_\theta/\partial z$ can depend differently on height, which difference is consistent with observations but contrary to the dictates of MOST above the SFL, in what Tennekes called the local free convection layer (Tenneekes, 1970). We will adopt the 2T model as a useful, qualitative guide as we seek to understand the mechanisms that underly the observed, ensemble-mean properties of temperature and heat transport in the surface layer.

### 4 Experimental

In this paper we focus on observations made during the SLTEST experiment, conducted in the Great Western Desert of Utah in 2005. The SLTEST site was close to an ideal site for our experiment. Nine sonic anemometers were mounted on a mast, facing North and spaced logarithmically over an 18-fold range of heights, from 1.43 m to 25.7 m. The fetch area was a flat, unobstructed playa surface stretching for 100 km to the North. The surface itself was particularly smooth during the experiment in May and June of 2005, since frequent rain in the winter and spring had eliminated the usual surface cracking and suppressed the growth of salt crystals, to give a very uniform surface of finely dispersed clay. This reduced surface drag and resulted in SFLs that were often only a few meters deep so, with winds predominantly from the North, the experiment yielded an unexpectedly large number of useable observations from above the SFL. On other occasions the SFL was deeper so the instruments recorded profiles within the SFL, as was envisaged in the original experimental design. Further information is given by McNaughton et al (2007).


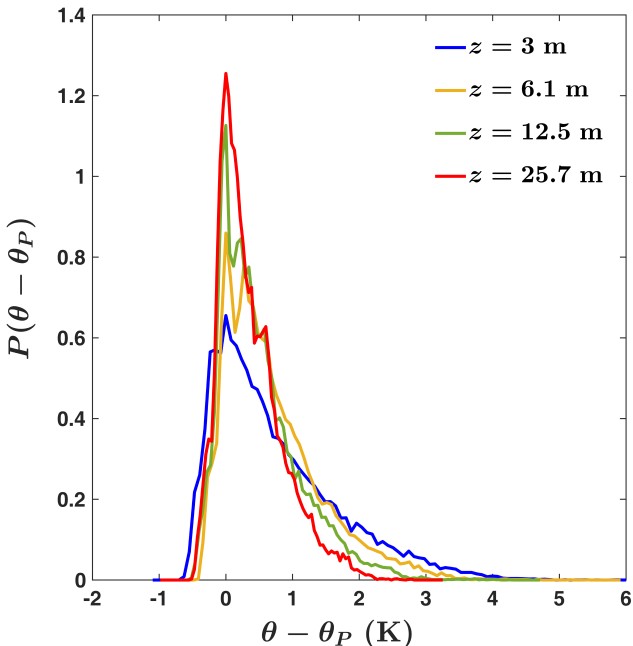

**Figure 2.** Probability distribution of temperatures at four heights above the SFL site at 2100-2130 on 24 May 2005, when $\lambda = 1230$ m. Here $\theta_P$ is the peak temperature. The distribution is strongly skewed towards the cooler temperatures of the air subsiding from the mixed layer, while the rising warm plumes occupy less area and so carry less weight in this representation.

## 5   Plumes above the SFL

The $T$ and $w$ signals contain a great deal more information than just their means, standard deviations and total heat fluxes, and this information can be used to explore the properties of plumes and heat-flux events. A probability density plot of temperature, for example, gives information on the relative areas occupied by plumes of various temperatures, while $T$ spectra and $wT$ cospectra give information on the lengths of plumes and flux events, and on their scaling properties. We now look at such information, starting with observations from above the SFL, where $u_\epsilon$ has no effect so our set of similarity parameter reduces to $\{z, \lambda, \epsilon_o, H\}$.

### 5.1   $T$ and $wT$ probability distributions

Temperature probability density functions (PDFs) directly represent the relative areas occupied by plumes of each temperature. Fig. 2 shows PDFs at four heights, all above the SFL during a particularly unstable run at SLTEST. These PDFs display a wide range of temperatures, and so depart strongly from the 2T model. Even so, we can use the idea of up- and down-plumes with fixed temperatures to interpret these curves. The prominent peaks at the four levels shown in Fig. 2 represent large areas of cooler air with a relatively small range of temperatures. That is, it represents air subsiding from near the top of the CBL. Up-



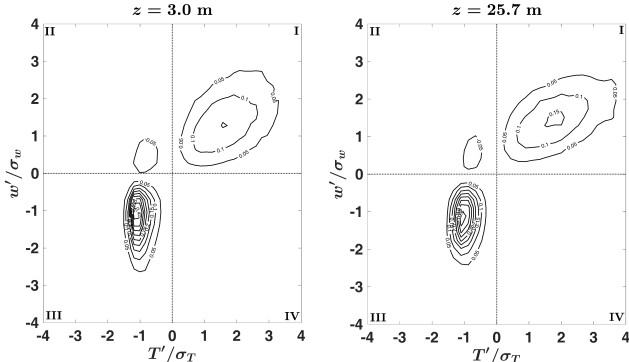

**Figure 3.** Joint $w'$-$T'$ distribution at 3.0 m and 25.7 m plotted on the $w'/\sigma_w, T'/\sigma_T$ plane and averaged over a selection of runs when both levels were above the SFL and 900 m $< \lambda <$ 1300 m. Heat transport is highly organized, being dominated by warm updrafts (I) and cool downdrafts (III) with little reverse transport in quadrants II and IV.

plumes are less-well represented on the probability plot because they occupy less area, and this smaller area decreases further with height, so probabilities of warmer temperatures falls away rapidly with $z$ while the fractional areas of the down-plumes increases. The peak heights therefore increase with $z$. The mean temperature also decreases with height.

Fig. 3 supports this interpretation. It shows the distribution of the heat flux on the $w'$ and $T'$ axes at 3.0 m and 25.7 m averaged over a selection of runs when both levels were above the SFL. It shows the joint probability distribution of $w'$ and $T'$ weighted by the product $w'T'$ at each point. The contributions of warm up-plumes (quadrant I) and cool-down plumes (quadrant III) are quite distinct, and very little heat is transported in the 'wrong' direction, in quadrants II and IV. Heat transport is therefore highly organized above the SFL.

### 5.2 $T$ spectra and $wT$ cospectra above the SFL

The mean profiles and PDFs considered so far tell us nothing about the sizes of plumes. For that we turn to the analysis of $T$ spectra and $wT$ cospectra. The surface layer above the SFL is a good place to begin because the power-law forms of the $T(z)$ and $\sigma_\theta$ profiles above the SFL imply self-similar properties for the underlying plumes there. McNaughton et al (2007); Chowdhuri et al (2019) have identified scales that collapse the $T$ spectra and $wT$ cospectra observed over a range of conditions onto 'universal' curves, so demonstrating self-similarity in the properties of the plumes that give rise to them. These scales provide information on what controls the behavior of the underlying plumes. In this they followed along the path pioneered by Perry and Abell (1975), who studied eddies using these techniques.

McNaughton et al (2007); Chowdhuri et al (2019) found that $T$ spectra and $wT$ cospectra observed above the SFL collapse onto universal curves in three wavenumber ranges, each with its own scales. We will call these the small-, mid- and large-wavenumber ranges. The mid- and large-wavenumber spectra appear to be universal across sites, while the small-wavenumber spectra are somewhat different at the very uniform SLTEST site and the more heterogeneous site of the CAIPEEX experiment



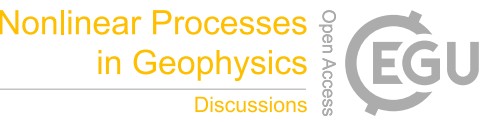

in India (Chowdhuri et al, 2019). This lack of universality among sites is consistent with the well-known differences observed for the largest flow structures in different flows, such as pipe, channel and boundary layer flows (Monty et al., 2009). It seems that the largest flow structures are influenced by the boundary conditions, particularly as they influence the overall flow geometry, while the mid- and small-scale structures depend on internal dynamics only and so have universal properties.

Here we review the scaling results of McNaughton et al (2007); Chowdhuri et al (2019) and extend our interpretation of
them. The scales found to collapse the $T$ spectra and $wT$ cospectra in each of the three ranges at SLTEST are given in Table 1. The spectra themselves are given in the original reports. A novel feature of the length scales is that many of them are mixed scales, or even doubly-mixed scales. Here we adopt a qualitative explanation for mixed scales based on our general understanding that eddies of all scales in the CBL are parts of a single, coordinated flow system. We take it that a mixed length scale $\ell_1^{1/2}\ell_2^{1/2}$ implies that a larger-scale eddy process, with length scale $\ell_1$, 'organizes' a smaller plume with length scale $\ell_2$.
Doubly-mixed length scales are interpreted similarly, with $\ell_1^{1/2}\ell_2^{1/2}$-scale eddies organizing $\ell_3$-scale plumes. We will apply this interpretation to the empirical scaling results.

### 5.2.1 Mid wavenumbers

We start with mid-wavenumber ranges of $T$ spectra and $wT$ cospectra because these are found to be universal across sites. The mid range includes the peak regions of these spectra and accounts for most of the $T$ variance and $wT$ fluxes observed
throughout the surface layer. More practically, the search for the scales able to collapse mid-range spectra is simpler because peak wavenumbers can be scaled unambiguously before going on to choose variance and covariance scales. These procedures are described by McNaughton et al (2007); Laubach and McNaughton (2009), and the identified scales are shown in Table 1.

A notable feature of the scales recorded in Table 1 is that many of them are mixed scales, so that the properties of spectra depend on the "outer" length scale $\lambda$ even quite near the ground. This is in good qualitative agreement with the DNS simulation
results of Fodor et al (2019), who found Monin-Obukhov similarity theory inadequate when describing plume properties and concluded that "updraft properties are not just determined locally, but also by outer scales". Our empirical studies of the scaling properties of spectra confirm that observation. We now turn to a more detailed interpretation of these scales.

The $T$ variance scale is $H^2(z\epsilon_o)^{-2/3}$, which implies a plume velocity scale $(z\epsilon_o)^{1/3}$. This is also the $w$ variance scale, and it identifies impinging outer Richardson eddies as the eddies active in shaping the plumes at this scale (McNaughton et al,
2007). The length scale here is the mixed length scale $\lambda^{1/2}z^{1/2}$, which we interpret as signifying an interaction between the plumes created by the impinging outer Richardson eddies and the larger, $\lambda$-scale eddies whose lower parts extend down into the surface layer. We need more information before we can go further, so we delay further interpretation until the end of this section. However, we do note that the peaks of the $T$ spectra lie at $\kappa\lambda^{1/2}z^{1/2} = 5.5$ (Chowdhuri et al, 2019), so peak plumes are long in the streamwise direction, and they elongate further as height increases. This is consistent with the sheet plumes that
comprise the rising walls of warm air that converge towards the bases of the thermals at the top of the surface layer.

At first sight the $T$ variance scale, $H^2(z\epsilon_o)^{-2/3}$, is incompatible with the 2T temperature scale, $(\theta_u - \theta_d)$, since the latter is independent of $z$. Reconciliation lies in the changing identity of the plumes above the SFL. Larger and larger impinging outer Richardson eddies sweep together more and more of the 'original' plumes rising from within the SFL, and in doing so they

**Table 1.** Length and variance scales of $T$ spectra and $wT$ cospectra measured above the surface friction layer. This table summarizes the scales identified by McNaughton et al (2007) and Chowdhuri et al (2019).

| Scales | Small wavenumbers | Mid wavenumbers | Large wavenumbers |
|---|---|---|---|
| $\kappa S_{ww}$ | | | |
| length | $\lambda$ | $z$ | $z$ |
| variance | $(\lambda\epsilon_o)^{2/3}\left(\frac{z}{\lambda}\right)^{4/3}$ | $(z\epsilon_o)^{2/3}$ | $(z\epsilon_o)^{2/3}$ |
| $\kappa S_{uu}$, $\kappa S_{vv}$ | | | |
| length | $\lambda$ | $\lambda$ | $\lambda$ |
| variance | $(\lambda\epsilon_o)^{2/3}$ | $(\lambda\epsilon_o)^{2/3}$ | $(\lambda\epsilon_o)^{2/3}$ |
| $\kappa S_{TT}$ | | | |
| length | $\lambda$ | $\lambda^{1/2}z^{1/2}$ | $\lambda^{1/4}z^{3/4}$ |
| variance | $H^2(\lambda\epsilon_o)^{-2/3}\left(\frac{z}{\lambda}\right)^{-1/3}$ | $H^2(z\epsilon_o)^{-2/3}$ | $H^2(z\epsilon_o)^{-2/3}$ |
| $\kappa S_{wT}$ | | | |
| length | $\lambda$ | $\lambda^{1/4}z^{3/4}$ | $z$ |
| covariance | $H\left(\frac{z}{\lambda}\right)^{1/2}$ | $H\left(\frac{z}{\lambda}\right)^{1/12}$ | $H$ |

laminate these plumes with more and more of the cooler air between them. As a result, the plumes whose lengths scale on $H^2(z\epsilon_o)^{-2/3}$ are composite plumes with characteristic temperatures that decrease with height. If we focus on just the original plumes, and if the identities of such plumes were unaffected by dissipation (as in the 2T model) then the temperatures of these plumes would remain constant with height.

The $T$ variance scale can also be written as $H(z_s\epsilon_o)^{-1/3}(z/z_s)^{-1/3}$, and as $H(\lambda\epsilon_o)^{-1/3}(z/\lambda)^{-1/3}$. The first of these is the temperature scale appropriate for plumes within the SFL, but modified by the height function $(z/z_s)^{-1/3}$ to express the diminishing the fraction of 'original' plume air as height increases. The second introduces the temperature scale of the thermals at the top of the surface layer, $H(\lambda\epsilon_o)^{-1/3}$ times the height function $(z/\lambda)^{-1/3}$, which represent the same aggregation process towards the final temperature of the thermals. This flexibility of interpretation reflects the scale-free nature of the process itself: the whole up-plumes are embedded in regions of the flow that converge laterally and accelerate upwards, and the shapes of





the embedded original plumes within the composites will be similar to the envelope shapes of the composite plumes. This

self-similar property is consistent with the length-scale-independent, power-law form of the mean temperature profile.

We can go further by focussing on the first interpretation: that the plume properties reflect the action of impinging outer Richardson eddies, which sweep together larger and larger aggregates of the original plumes rising from within the SFL. If these impinging outer Richardson eddies acted in isolation from eddies of other kinds then plume lengths would scale on $z$ alone, not on $\lambda^{1/2}z^{1/2}$. The difference can be attributed to the up-plumes all being embedded in regions of the flow where the

horizontal convergence depends on $(z/\lambda)^{-1/2}$. The absence of heat flux in quadrant IV of Fig. 3 tells us that the up-plumes are all associated with the updrafts created by these convergences, so we can directly associate the $(z/\lambda)^{-1/2}$ power law of the convergences with a similar dependence of the total area of the up-plumes. The length scale displayed by the mid-range of the $T$ spectrum is consistent with the $-1/2$-power law followed by the mean temperature profile.

Scales for $wT$ cospectra are more complicated. The lengths of $wT$ flux events scale on $\lambda^{1/4}z^{3/4}$, which is a doubly-mixed

scale (Table 1). It indicates that the structures contributing to the flux depend on an interaction between attached eddies with mixed length scale $\lambda^{1/2}z^{1/2}$ and up-plumes of length scale $z$. Length $z$ occurs twice here, but we recall that impinging outer Richardson eddies can have a wide range of sizes, so plumes created by smaller impinging outer Richardson eddies can be acted on by larger impinging outer Richardson eddies in a self-similar fashion over a range of heights. That the smaller impinging outer Richardson eddies can themselves be characterized by $z$ alone, even while embedded in a larger-scale flow that bears

the $\lambda$ scale, is confirmed by the absence of mixed scales for the velocity spectra in Table 1. We infer that the parameter $z$ plays two roles: the first in raising plumes with scale $z$ from near the ground, and the second in aggregating those plumes into $z^{1/2}\lambda^{1/2}$-scale composites created by the flow fields of larger impinging outer Richardson eddies being modified by their embedding in the $\lambda$-scale flow field of the large eddies.. We have found no explanation for the surprising $(z/\lambda)^{1/12}$ dependence of the covariance scale, though it is consistent with preserving flux over the whole cospectrum (Chowdhuri et al, 2019).

Since $T$ spectra and $wT$ cospectra have different length scales it follows their peaks are not at the same wavenumber. The peaks for $wT$ cospectra are found to be at $\kappa\lambda^{1/4}z^{3/4} = 1.2$ while those of the $T$ spectra are at $\kappa\lambda^{1/2}z^{1/2} = 5.5$ (Chowdhuri et al, 2019). Peak wavenumbers for $wT$ flux events are therefore from 1.5 to 2.5 times larger than those for $T$ plumes as $z/\lambda$ increases from 0.01 to 0.1. That is, flux events are shorter than plumes, so we must conclude that heat transport is concentrated in the faster-rising parts of plumes. A plausible explanation is that the faster-rising parts of up-plumes lie closer to the roots of

the thermals where uplift is strongest.

### 5.3 Small and large wavenumbers

The mid-range plumes aggregate as height increases, eventually to combine into the $\lambda$-scale clusters we know as thermals. Between these lie cooler areas of descending air with smaller temperature fluctuations, and these areas increase with height. As a result the Fourier representations of temperature signals change with height, becoming richer in small-wavenumber com-

ponents but poorer in mid-range components as height increases. The same is true of $wT$ cospectra, so heat flux is transferred from the mid-ranges of $wT$ cospectra to their small-wavenumber ranges as height increases. The small-wavenumber parts of spectra and cospectra collapse when wavenumbers are scaled on $\lambda$ (Table 1).





The mean temperature gradient is insensitive to these changes in spectral representation. The same original plumes still taper in the same way, so the combined area of these up-plumes continue to follow the $z^{-1/2}$ power law, even while their spectral representation is sensitive to their changing aggregation properties. In the small-wavenumber range the covariance increases in proportion to the $(z/\lambda)^{1/2}$ increase at each wavenumber, which $1/2$ power suggests a link to the $(z/\lambda)^{-1/2}$ decrease in the envelope areas of clusters of original up-plumes. We have no formal analysis to confirm this connection.

The large-wavenumber ranges of spectra and cospectra reflect the action of detached outer Richardson eddies on the mid-range plumes, as indicated by the doubly-mixed length scale of the $T$ spectrum in this range. This combines the $\lambda^{1/2}z^{1/2}$ length scale of the plumes with the $z$ length scale of the outer Richardson eddies. They carry this scale even while detached because their population still reflects the influence of the ground. The Richardson cascade is itself scale-free, and characterized by the $-5/3$ power-law Komogorov spectrum. However, this spectrum is also characterized by the absence of eddies with heights significantly greater than $z$, because such large eddies increasingly interact with the ground (become attached) and so behave differently. Detached outer Richardson eddies create fine structure in plumes by stretching and folding them repeatedly, especially at their margins, eventually creating plume filaments at scales where molecular dissipation becomes important.

Detached outer Richardson eddies have no intrinsic sense of up and down, so the direction of heat transport is set by the mean temperature gradient. Such transport does not change the temperature profile itself. We have a scale-free population of detached Richardson eddies acting on a scale-free temperature profile, so the resulting heat flux is scale free, and so independent of height. This independence is confirmed by observations, which show that $wT$ cospectra collapse over a range of heights when scaled on $H$ (Table 1). With no flux divergence the mean temperature profile is unaffected. This model of random transport down a pre-existing temperature gradient is consistent with the 'eddy diffusion' model of Wyngaard and Coté (1972), which predicts that the $wT$ cospectrum should follow a $-4/3$ power-law in this range, as is observed (e.g. Chowdhuri et al, 2019).

## 6 Plumes within the SFL

Plumes within the SFL are more complicated. We know that the power-law form of the mean temperature gradient extends down into the SFL, but follows a log law very near the ground. We therefore do not expect plumes to have a single, self-similar form throughout the SFL, but one that changes with height. The nature of that change is our focus here.

Though many experiments have been conducted within the SFL, most published results are of limited value for our work. Only the results of Laubach and McNaughton (2009), have been reported using non-dimensionalizations based our preferred parameter set, $\{\lambda, z, \epsilon_o, u_\epsilon, H\}$; and even those results are problematic because they were from experiments not originally designed for our kind of analysis. Here we base our interpretations on results from a new analysis of the SLTEST data. We first describe how data were selected, then introduce the new results as we go along.





## 6.1 Data selection and processing

A difficulty encountered when using our parameter set is that calculation of the outer dissipation rate, $\epsilon_o$ must be based on observations made well above the SFL, often at many tens of meters above ground, while the dissipation velocity, $u_\epsilon$ must be based on observations made very near the ground, at small $z/z_s$. That is, instruments must be mounted at two widely-spaced heights if both parameters are to be determined directly. The SLTEST experiment was designed with this in mind, and instruments were arranged so that they would span the SFL on many occasions. Laubach and McNaughton (2009) used a limited range of instrument heights—just one at their principal site—so $z/z_s$ was essentially an inverse scale variable in their analysis, while $z_s$ was calculated using an empirical relationship for $\epsilon_o$ taken from KY90. We have repeated their analysis using SLTEST data.

For the present analysis we first selected all daytime runs from SLTEST where wind was from Northerly directions, and for each of these we calculated trial values of $\epsilon_o$ and $u_\epsilon$ by assuming that the highest sonic anemometer lay well above the top of the SFL and the lowest very near its bottom, and so calculated trial values of $z_s$ for each run. Runs were accepted for further analysis when 7.1 m $< z_s <$ 12.8 m, which values of $z_s$ validate our assumptions that the top sonic anemometer was at $z/z_s > 2$ and the bottom one at $z/z_s < 0.2$. By this means 32 runs were selected for further analysis, and for each of these the sonic anemometers were well spread across the SFL. This makes $z/z_s$ a true scaled height variable in our analysis. A limitation of the SLTEST results is that even an 18-fold range of anemometer heights is rather small, so we have a rather narrow range of $z_s$ values to work with. While some of the SLTEST results differ those reported by Laubach and McNaughton (2009) some also agree, as we point out. The earlier results are still very valuable, being from two experiments where $z_s$ varied widely even while having a limited range of $z$ values.

We present our spectral results on wavenumber axes rather than frequency axes even while the basic data are from time-series observations made by fixed instruments. This choice is to facilitate interpretation in terms of the geometries of eddies and plumes. We adopt Taylor's frozen turbulence hypothesis and use mean wind speed at each level to make the conversion from frequency to wavenumber, in the standard way. In earlier papers we performed similar analyses of specta above the SFL, and for these we used the single mean velocity at the top of the mast to make the conversion at all levels, noting that wind speed varies only slowly with height above the SFL (McNaughton et al, 2007; Chowdhuri et al, 2019). Neither procedure is ideal since plumes of different sizes move at different speeds in sheared atmospheric surface layers (Davison, 1974; Cheng et al, 2017). Use of Taylor's hypothesis leads to compression of the wavenumber axis of the $T$ spectra and $wT$ cospectra. Even so, the scales that collapse these spectra and cospectra are unaffected, and so too are our interpretations of those scales.

## 6.2 $T$ and $wT$ probability distributions

Fig. 4 shows the $T'$ probability distribution at SLTEST at four height ranges, calculated for just the fluctuations in the mid-range of the spectrum, obtained by first low-pass-filtering the signal with cut-off wavenumber $\kappa z = 1$. Fig. 5 shows the joint distribution of the mid-range heat flux mapped on $w'/\sigma_w$ and $T'/\sigma_T$ axes. Starting with Figs. 2 and 3, which show results from above the SFL, then moving progressively down through the SFL in Figs. 4 and 5, we see a continuous progression

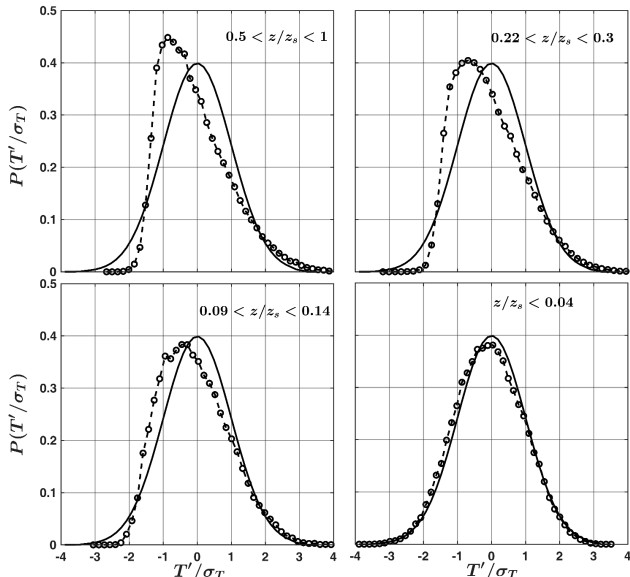

**Figure 4.** Temperature probability distributions at four heights at SLTEST. Temperature signals were first filtered using a Fourier low-pass filter set at $\kappa z = 1$ to remove high-frequency fluctuations. Data are from the 32 runs whose selection is described in the text, except that $\epsilon_o$, and so $z_s$ could not be measured directly for runs where $z/z_s < 0.04$. Runs for this case were selected by assuming $z_s = -L$, and further selected so that $-L < 100$ m to ensure that the flow regime was always CBL.

from well-organized transport above the SFL to near-Gaussian behavior at its bottom. It seems that plume temperatures have a Gaussian distribution at their source at the ground, in strong contrast to the 2T model, and that this initial distribution is generally maintained as the up-plumes rise. The temperature distribution of the down-plumes can be interpreted as reflecting the combination of a decreasing area fraction of original down-plume air, and its mixing with up-plumes having a Gaussian temperature distribution, with the vertical velocities of the composites becoming increasingly random as height diminishes. (We recall that the up- and down-plume terminology is based on the origin of the plume, not the current direction of movement.) This progression of plume properties is generally consistent with expectation based on the mean profiles, but to go further we must assemble more evidence. We now turn to the $T$ spectra and $wT$ cospectra.

### 6.3 $T$ spectra and $wT$ cospectra within the SFL

$T$ spectra for the 32 selected runs from SLTEST are shown in Fig. 6. The peak positions collapse when lengths are scaled on the doubly-mixed scale $\lambda^{1/4} z_s^{1/4} z^{1/2}$, in agreement with the collapse shown in Laubach and McNaughton (2009, Fig. 10). This behavior is therefore consistent across three experimental sites. Fig. 6 also shows that spectrum heights collapse at all wavenumbers in most of the SFL when variance is scaled on $H^2 (z\epsilon_o)^{-2/3}$, in agreement with the $z^{1/3}$ power law for the $\sigma_\theta$ profile described above. The same scale collapsed the mid ranges of $T$ spectra above the SFL (Table 1). Only the lowest level





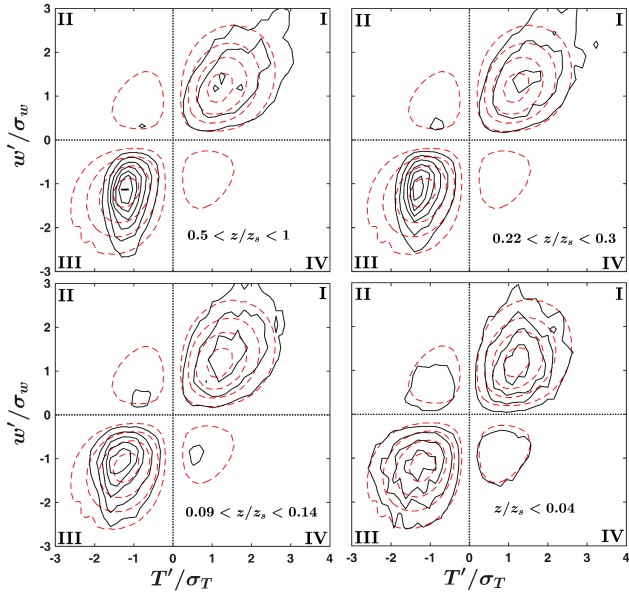

**Figure 5.** Accumulated heat fluxes, $w'T'/H$ mapped onto the $(w'/\sigma_w, T'/\sigma_T)$ plane. The red contours represent two Gaussian signals with the same flux. Run details are as for Fig. 4.

shown in Fig. 6, for $0.09 < z/z_s < 0.14$, deviates significantly. That is, plume self-similarity does not extend right down to the ground.

We note that plume length scales are different above and within the SFL: the appropriate scale above the SFL being the mixed scale $\lambda^{1/2}z^{1/2}$ (Table 1), but the doubly-mixed length scale $\lambda^{1/4}z_s^{1/4}z^{1/2}$ within it. These scales both depend on $z^{1/2}$ and match at $z = z_s$, so profiles are continuous there. We have already interpreted the mixed scale above the SFL as reflecting

the scale of plumes created by the action of $z$-scale impinging outer Richardson eddies. The consistent explanation for the doubly-mixed scale is that we have plumes created by the action of $z$-scale impinging outer Richardson eddies, but embedded within and shaped by the converging flow field of the larger and more powerful, $\lambda^{1/2}z_s^{1/2}$-scale eddies. If so the impinging outer Richardson eddies that act on the $z$-scale plumes have a fixed length at all heights within the SFL. This is consistent with our conceptual model of the SFL, in which the only impinging outer Richardson eddies able to penetrate the SFL are those

large enough to survive direct interaction with the more-powerful shear eddies within the SFL.

An interesting feature of our results is the small shoulders lying a decade and more to the right of the main peaks in both $T$ spectra and, more prominently, in the $wT$ cospectra shown in Fig. 7. These shoulders indicate emerging peaks that become more prominent near the ground and lie at the beginnings of the large-wavenumber parts of the $wT$ cospectra. Fig. 7 shows the same cospectra in both panels, but with wavenumbers scaled in two different ways. The doubly-mixed length scale $\lambda^{1/4}z_s^{1/4}z^{1/2}$

gives better collapse of the positions of the main peaks, but simple $z$ scaling gives better collapse of the subsidiary peaks. At larger wavenumbers the whole $wT$ cospectrum scales on $z$. We also notice that the main peak diminishes with height as


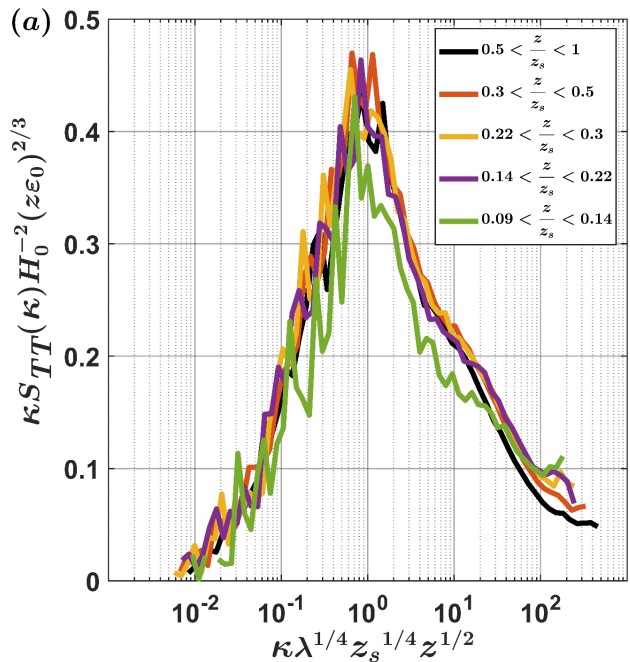

**Figure 6.** $T$ spectra from within the SFL at the SLTEST site. The variance scale is that which collapses peak heights above the SFL, though the length scale is now doubly-mixed. The scaled spectrum begins to change only at the lowest levels, $z/z_s < 0.14$.

the minor peak grows. All of these features support our interpretation that two kinds of eddies, and so two kinds of plumes, transport heat within the SFL. These finding agree with those of Laubach and McNaughton (2009, Fig. 12), so the results are again consistent across sites.

We note that Smedman et al. (2007) also reported $wT$ cospectra with two peaks, both in positions consistent with our peaks, but at small $-z/L$ their larger-wavenumber peak is much bigger than that shown in Fig. 7. They interpret its emergence as indicating a transition to a distinct Unstable-Very-Close-to-Neutral (UVCN) flow regime. We agree, but note that our runs are selected so as to avoid sampling this near-neutral regime. The defining characteristic of a CBL regime is the presence of an outer layer deep enough to support an outer Richardson cascade. The thickness of this outer layer reduces as the heat flux
decreases because the SFL deepens as neutrality is approached. The remaining outer layer, of depth $(z_i - z_s)$, must eventually become too shallow to accommodate an outer Richardson cascade, whereupon the CBL regime will transition to a near-neutral-boundary-layer (NNBL) regime. We use NNBL rather than UVCN to describe this regime since there is evidence that it persists into slightly stable conditions, where it is described as the 'strong turbulence regime' of stable boundary layers. Smedman et al. (2007) had no direct way to discriminate CBL and NNBL regimes, so their results reflect a mixture of both. Laubach and
McNaughton (2009) found a similar $z$-scaled peak emerged at small $-z/L$, but their observation height was fixed so this always occurred in near-neutral conditions, and NNBL results were not excluded.

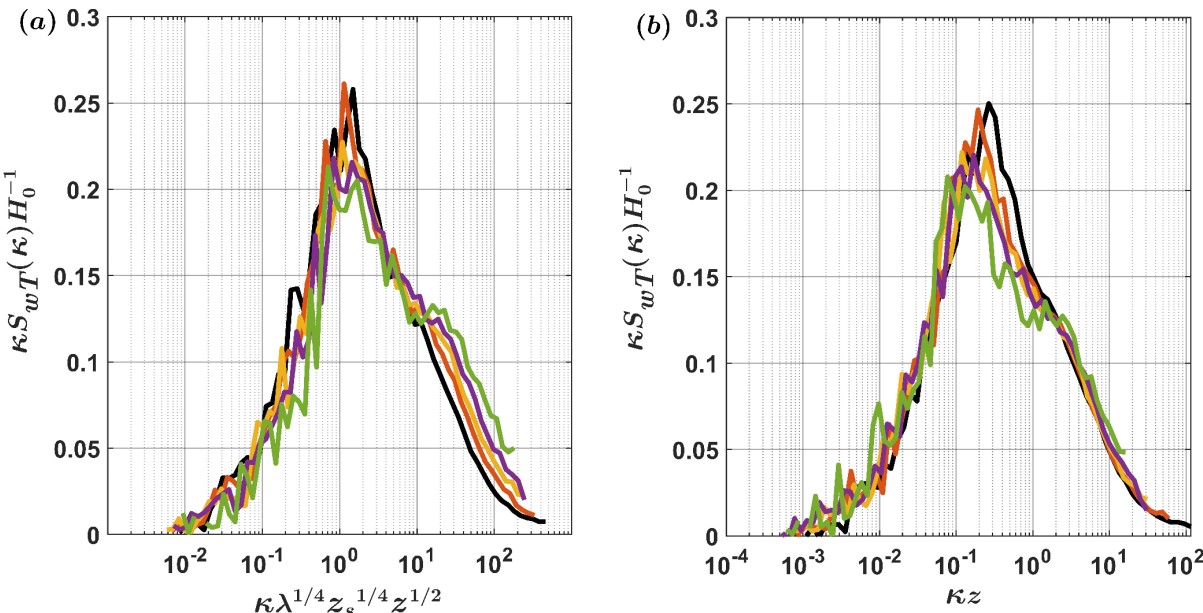

**Figure 7.** $wT$ cospectra from within the SFL at the SLTEST site. The doubly-mixed plume length scale in (a) is chosen to collapse the positions of the main peaks, while in (b) the simple mixed scale $z$ is chosen to collapse the position of the emerging minor peak at larger wavenumbers.

## 6.4 A conceptual model for plumes within the SFL

The parameterization of CBLs adopted in this paper is based on an idea introduced by McNaughton (2004). It is that there exists a special kind of eddies, called shear eddies, whose fundamental role is to transport momentum to the ground at small
heights where blocking by the ground limits the vertical motions of, and so the transport effectiveness of, larger eddies. These shear eddies are attached to the ground and their heights scale on $z$, so momentum can be carried right down to the ground by a sequence of ever-smaller shear eddies. That this process is rather inefficient is shown by the high dissipation rates observed near the ground. McNaughton (2004) proposed that these shear eddies are initiated at the ground and grow upwards by a cascade process until their growth is disrupted at height $z_s$ by interaction with outer Richardson eddies. This interaction defines the top
of the SFL, and it places a lower limit on the size of the outer Richardson eddies that can penetrate the SFL so these eddies must become ineffective in transporting momentum as $z/z_s \rightarrow 0$.

The shear eddies also transport heat, but temperature is locally conserved so plumes have continuous identities while higher-momentum air parcels do not. We can imagine a first generation of up-plumes originating at some arbitrary but small height, and we can notionally track these up-plumes. Since the heights of the shear eddies are limited, being $\sim 2z$, a sequence of larger
and larger shear eddies must act on these plumes as they rise through the SFL. This $z$-scale process then creates the minor, $z$-scaled peak in the $wT$ cospectrum (Fig. 7), prominent only in the lowest part of the SFL.



We can then track these plumes, notionally at least, as they are swept together into composite plumes by the action of the larger, $z_s$-scale impinging outer Richardson eddies. The sheet plumes are formed by lateral flow convergences, which sweep the up-plumes together into the larger, composite structures we call sheet plumes, and accelerate them upwards. In this way the

up-plumes progressively acquire velocities related to the velocity scale of the impinging outer Richardson eddies, $(z_s \epsilon_o)^{1/3}$, rather than the velocity scale of the shear eddies, $u_\epsilon$, which was important in their creation and early rise. The shear eddies become disorganized and ineffective by the top of the SFL, so the velocity scale $(z_s \epsilon_o)^{1/3}$ prevails. The organizing effect of the $\lambda^{1/2} z_s^{1/2}$-scale impinging outer Richardson eddies on the $z$-scale 'original' plumes is also apparent throughout the SFL, as shown by the doubly-mixed length scale for the composite plumes in the same spectrum.

The $T$ and $wT$ probability distributions support this interpretation. The temperature distribution is approximately Gaussian near the ground, as expected when plumes are created at very small scale where molecular diffusion is important, and are transported upwards by what is effectively a random walk process whose scale increases in proportion to height. This near-Gaussian temperature distribution then persists in the up-plumes as scale increases, molecular diffusion become less important, and plume identities become more fixed. This can be seen in the up-plumes shown in the joint $wT$ distributions in Fig. 5. The

down-plumes, on the other hand, are rather distinct at the top of the SFL, but are increasingly stretched, folded and entwined with the up-plumes as the ground is approached, so that finally up- and down-plumes lose their separate identities and combine to give a single Gaussian temperature distribution.

That is to say, heat transport by shear eddies is effectively an 'eddy diffusion' process, similar to that assumed in traditional mixing length models. In confirmation of this the large-wavenumber part of the $wT$ cospectrum (Fig. 7) follows a $(\kappa z)^{-4/3}$

power law when plotted on log-log axes, consistent with the gradient-diffusion assumption of Wyngaard and Coté (1972), though here the shear eddies create the randomness. The impinging outer Richardson eddies aggregate these original plumes and impose increasing order with height. This is shown in Figs. 4 and 5, where the reverse transport in quadrants II and IV diminishes progressively with height.

Another feature of the SFL is that the mid-ranges of $T$ spectra and $wT$ cospectra share the same length scale, $\lambda^{1/4} z_s^{1/4} z^{1/2}$,

and indeed the peaks are found at the same wavenumbers. That is, heat is transported uniformly along the plumes, whether because plumes of uniform temperature rise at a uniform velocity along their length or because their vertical velocities and temperatures are inversely related: the air being hotter where the plumes rise less rapidly. This contrasts with the situation above the SFL, but is consistent with a uniform forcing by the heat flux at the ground.

Plume lengths at the top of the SFL necessarily match with those immediately above it, but they scale differently since within

the SFL the impinging outer Richardson eddies have a minimum size which scales on $z_s$. Even so, both length scales depend on $z^{1/2}$ so the temperature gradient is continuous down through the top of the SFL, and this gradient will continue on down so long as the impinging outer Richardson eddies play the dominant role in determining the areas of the rising plumes. Fig. 1 shows that the $z^{-1/2}$ temperature profile continues down to $-z/L \lesssim 0.4$. An even lower transition level is consistent with the $T$ distributions and joint $wT$ distributions shown in Figs. 4 and 5. Indeed the $T$ spectra in Fig. 6 show that the transition is

only just beginning at the lowest levels available for analysis, $0.09 < z/z_s < 0.14$. A very low transition in the temperature gradient is consistent with the findings of Priestley (1955); Foken and Skeib (1983), both of whose instruments spanned more



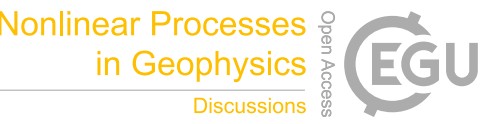

than a 20-fold range of heights. Priestley (1955) found the log law to apply only up to $-Ri = 0.03$, where $Ri$ is the gradient Richardson number, while Foken and Skeib (1983) found it to apply up to $-z/L = 0.06$. This profile is very different to that reported by Businger et al (1971), whose instruments spanned a much narrower range of heights.

## 7 General Discussion

This paper has given an account of the scaling properties of temperature profiles and plumes, and of $T$ spectra and $wT$ cospectra in terms of known properties of the eddies and plumes that comprise turbulent flow in the surface layer of convective boundary layers. The 2T model has contributed to this by allowing us to talk of the geometric properties of plumes, even if only by analogy with idealized plumes in a system where plumes can be strictly defined and where temperature variance is conserved. The distinction between $\sigma_{<w>}$ and $\sigma_w$ is important in this interpretation, particularly within the upper SFL where $\sigma_w$ scales on $u_\epsilon^2$ while $\sigma_{<w>}$ has no simple scale because shear and impinging outer Richardson eddies both contribute to plume velocities there.

### 7.1 The role of buoyancy

Our account is based on concepts that are, in some respects, foreign to the statistical fluid mechanics (SFM) understanding of turbulence in CBLs. In particular, buoyancy plays a very different role in our conceptual model of convection. To explain how such a different approach can be possible we first review the origins of the SFM model, which model encompasses the work of Reynolds (1895) and Richardson (1920), who laid down the concepts, Monin and Obukhov (1954) who formalized them in a useful way, and many later authors who have developed empirical relationships based on these concepts.

The origins of SFM lie in the work of Reynolds (1895) who, given the intractability of the Navier-Stokes equations, decided that turbulence could only be addressed statistically. He developed the set of Reynolds-averaged Navier-Stokes (RANS) equations, but formal rigor in this development was achieved much later, by Kolmogorov. The problem was that Reynolds' volume averages could not properly be defined in flows whose largest eddies span the flow (McNaughton, 2012). Kolmogorov solved this by redefining the volume averages as ensemble averages, but in this formulation the RANS equations can contain no information on the dynamics of flows. To form an ensemble average one takes a series of 'snapshots' of a flow, either from separate flow realizations (experiments or simulations) or from a single flow but separated widely enough in time to ensure no statistical connection between the snapshots. This very procedure removes all dynamical information since dynamics is about causal connections between successive states of the flow. Ensemble averaging also smears out any transient spatial patterns in the flow so that the forms of eddies and plumes can not be represented by ensemble-averages. Much recent work has attempted to re-introduce information on flow structures by melding ideas from SFM with ideas taken from the study of physical flow structures, but we have taken an independent path based only on ideas taken from the study of complex dynamical systems (CDS).

Our conceptual departure from SFM is most noticeable in our treatment of buoyancy. Richardson (1920) extended the RANS equations by adding a buoyancy term to Reynolds' original equations for neutral flows. A detailed account of Richardson's





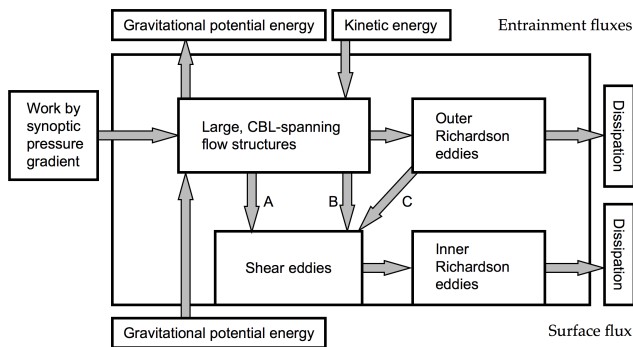

**Figure 8.** The flow of mechanical energy in CBLs. Arrows represent energy transfers from one kind of eddy to the next. The energy flowing from the large, CBL spanning structures to the shear eddies has been divided into three streams, A, B and C. The first of these supports the mean transfer of shear stress to the ground while the second and third supports the fluctuations. The relative sizes of these flows change with instability. Adapted from McNaughton (2012).

arguments are beyond the scope of this discussion, but he concluded that turbulence kinetic energy is increased by the local

action of buoyancy forces which accelerate warmer parcels of air upwards, and that this 'buoyant production' was particularly important near the ground where temperature gradients are largest. This interpretation has persisted (e.g. Monin and Yaglom, 1971; Wyngaard, 2010) despite the impossibility of any such dynamical interpretation once ensemble averages were adopted. In fact, the RANS energy equation simply expresses a balance between the divergence of the flux of mechanical energy and the local dissipation rate, and it can be derived from that principle alone, without reference to the Navier-Stokes equations.

Though the magnitude of the 'buoyant production' term in the RANS energy equation can be significant, this does not signify any particular action of buoyancy forces.

In CBLs mechanical energy flows into the system at the largest scale, passes down progressively, from one kind of flow structure to the next, to be finally dissipated as heat at the smallest scale, as shown in Fig. 8. The largest flow structures have the greatest energy densities and the longest lifetimes, so smaller eddies and smaller plumes are simply swept along, moving

within them in whatever direction and at whatever velocity the larger eddies dictate. In particular, plumes do not accelerate upwards under the action of local buoyancy forces. It is, however, intrinsic to the system-wide organization of the flow that the largest eddies accommodate the buoyancy of smaller plumes by aggregating them and incorporating them into the largest-scale patterns in the flow.

## 7.2 Flow of mechanical energy in CBLs

The action of buoyancy can be discussed in terms of the flows of mechanical energy in CBLs since a consequence of this large-scale organization of CBL flows is that the effects of the input of gravitational potential energy at the ground is expressed not by the smallest eddies near the ground, as Richardson (1920) would have it, but by the large coherent structures, and not




by causing plumes of warm air to accelerate upwards through the lower CBL, but by maintaining the balance between the re-formation and decay of the large coherent structures in the flow. The large structures then pass their energy down continuously

to smaller and smaller structures in which the form of the source eddies quickly becomes irrelevant. We can then parameterize the energies of the smaller eddies simply in terms of the energy flowing through each part of the system, as shown in Fig. 8. In our scaling scheme the buoyancy effects of the heat flux are therefore accounted through the dissipation rates, as stated in the introduction. This perspective is consistent with the importance of the dissipative flux in the theory of far-from-equilibrium systems (Evans and Morriss, 2007).

Fig. 8 is based on the diagram drawn by McNaughton (2012), but here the large, semi-permanent coherent structures and the largest turbulent eddies are combined and described as large, CBL-spanning flow structures. This change acknowledges the importance of the changing character of the large eddies with changing instability, as described by Jayaraman and Brasseur (2018). An extra energy flow pathway has been added so the mean and fluctuating effects of these large flow structures can be represented separately. Energy now flows to the shear eddies by three pathways, marked A, B and C in Fig. 8. The first

component, A, supports the transfer of mean shear stress to the ground while component B supports the fluctuations in shear stress created by the large coherent structures, and component C supports the smaller-scale fluctuations in surface shear stress created by outer Richardson eddies impinging onto the ground. All three will contribute to dissipation near the ground, and so to the value of $u_\epsilon$. Taken alone, an increase in $(F_A + F_B + F_C)$ will therefore cause an increase in $u_\epsilon$ and so an increase in $z_s$, where the $F$s are the fluxes of mechanical energy. However, we know that the form of the large CBL-spanning flow structures

do change with stability—from long roll vortices parallel to the wind in near-neutral conditions to cellular convective cells when the CBL is very unstable—so pathways B and C increase in importance as instability increases. The sum of these two energy flows is just $z_i \epsilon_o$, where $z_i$ is the height of the CBL, since the velocity scale of the large eddies can be parameterized as $(z_i \epsilon_o)^{2/3}$ and $\epsilon_o$ also parameterizes energy flow down the outer Richardson cascade (McNaughton et al, 2007). The effect is that the ratio $(F_B + F_C)/(F_A + F_B + F_C)$ increases as the large CBL-spanning flow structures become more cellular, so $z_s$

decreases as this ratio increases. Indeed, the ratio acts as a stability parameter for CBLs.

### 7.3 Stability parameters

In our work $z/z_s$ plays a role somewhat analogous to the role of $-z/L$ in MOST, and indeed the two are correlated (Chowdhuri et al, 2019). We can use Fig. 8 to explore the source of this correlation since both lengths can be derived from this scheme. As we have seen $z_s$ is given by (1), where $u_\epsilon$ depends on the mechanical energy dissipated within the SFL, and so on the sum

of the energies flowing down pathways A, B, and C. To calculate $L$ we can again use (1), but now we first delete the energy flow pathways B and C. Then the dissipation velocity, $u_\epsilon$, becomes $u_*$, and without pathways B and C all of the gravitational energy passes directly to outer dissipation, so $gH/T_0$ replaces $\epsilon_o$. This is so because buoyancy forces act vertically so none of their energy can go to supporting the mean momentum flux. With these changes $z_s = -L$. Though this leads us to expect good correlation between $z_s$ and $L$, it also leads us to expect that the Monin-Obukhov model will break down in windless

convection, when $u_*$ goes to zero. Several adjustments to the standard Monin-Obukhov model have been proposed to avoid this. Businger (1973) proposed that $u_*$ should have a lower limit $w_*$, where $w_*$ is the Deardorff convective velocity scale.





Beljaars (1994) redefined $L$ using $u_*^2 + \beta w_*^2$ in place of $u_*^2$, so adding a proxy for the missing energy flow in all conditions. Others, (e.g. Stull, 1988), have redefined $u_*^2$ as $(\overline{u'w'}^2 + \overline{v'w'}^2)^{1/2}$, which is clearly an *ad hoc* adjustment since an ensemble-averaged $\overline{v'w'}$ must be zero, but in practice this also seems to compensate for the missing energy flow (e.g. Wilson, 2008).

The other change—setting $\epsilon_o$ equal to $gH/T_0$—has some empirical support in the work of Kader and Yaglom (1990), but their empirical relationship, equivalent to $\epsilon_o = 1.1\, gH/T_0$, implies that the contribution of entrained kinetic energy to $\epsilon_o$ outweighs work done against buoyancy in the capping inversion in moderately and strongly convective conditions. This is disturbing in itself, and it must fail in less convective conditions when gravitational potential energy becomes the minor contributor to the outer budget of mechanical energy. Of the two relationships necessary to support a correlation between $z_s$ and $-L$, we expect

the $u_\epsilon = u_*$ approximation to become more accurate in nearer-neutral conditions but the $\epsilon_o = gH/T_0$ approximation to become worse.

## 8   Conclusions

In this paper we have surveyed ensemble-averaged observations of temperature profiles, $T$ spectra and $wT$ cospectra and probability plots from within CBLs, and given interpretations of them based on the conceptual model of eddy and plume

processes previously introduced by McNaughton et al (2007); Laubach and McNaughton (2009). This survey has been wider than the previous accounts, and it includes number of new results and interpretations. We have been able to identify the $z^{-1/2}$ power law for the mean temperature gradients above the SFL with the narrowing of rising warm plumes as they rise using a two-temperature (2T) toy model, then to connect this power law with the $\lambda^{1/2}z^{1/2}$ mixed length scale that collapses the positions of the peaks of $T$ spectra above the SFL. We have also presented results from a new analysis of $T$ spectra from within

the SFL at the SLTEST experimental site. These confirm some of the results reported by Laubach and McNaughton (2009), and they support a model where one kind of plume, created by the action of shear eddies, predominate only very near the ground ($z/z_s \lesssim 0.1$), but that these are soon modified by the $z_s$-scale impinging outer Richardson eddies in which they are embedded, which embedding dictates their areas and so the form of the temperature profile, even quite close to the ground. Quadrant plots of joint $w$-$T$ distributions support this interpretation.

A key feature of our survey is that it uses concepts that are consistent with our general understanding of the CBL as a dissipative complex dynamical system. In particular, we identify eddies and plumes as emergent structures that embody the high levels of organization. Emergence is a defining characteristic of all far-from-equilibrium, dissipative systems, whose high level of organization is sustained by a constant flow of energy. This energy takes the form of mechanical energy in dynamical systems, and our similarity scheme is based on the parameters that characterize this energy flow. The same perspective leads

to an understanding of buoyancy in CBL flows that is very different to that introduced a century ago, by Richardson (1920).

Even so, the new model raises many questions. We still cannot say why mixed scales always involve $1/2$ powers. We need a new experiment to characterize the temperature profile using the new similarity parameters. We do not know how to model entrainment in a way that will connect energy flows within the CBL to the synoptic-scale flow. This task of parameterizing entrainment is made more complicated by the expectation that the entrainment rate will depend on both the form and the energy





of the large structures within the CBL. These are matters for the future. Even so, we are optimistic that the present approach can be extended usefully to the near-neutral and stable regimes in atmospheric boundary layers.

*Competing interests.* We have no competing interests

*Acknowledgements.* We thank Nelson Dias and Johannes Laubach for helpful comments on earlier drafts of this manuscript.





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
