# Peer review of "Temperature profiles, plumes and spectra in the surface layers of convective atmospheric boundary layers"

_Nonlinear Processes in Geophysics, 2019_

## Referee Comment (RC1) · Arun Ramanathan (Referee) · 14 Oct 2019

**General Comments:** This paper deals with an important geophysical issue, namely the scaling properties of plumes within and above the surface friction layer of convective atmospheric boundary layers, and therefore suits the scope of Nonlinear Processes in Geophysics (NPG) journal. Although the present version of this manuscript is mostly well-written, it does lack meticulousness in quite a few places but can be accepted for publication if the authors take into account the following comments and make the corresponding changes. Furthermore, the authors must consider the fact that the readers of NPG are not limited to the field of boundary layer meteorology

while making these changes.

**Detailed Comments:** Changes corresponding to the following comments need to be incorporated

1. Please use continuous line numbers.

2. Page 1 Line 10: Introduce vertical velocity similar to the way temperature is introduced (in the same line rather than in the next line).

3. Page 1 Line 15: plumes above the surface friction layer (SFL).

4. Page 1 Line 15: state that $z, z_s$ are the observation and SFL heights respectively.

5. Page 1 Line 25: explicitly (but briefly) mention this contrasting interpretation of the role of buoyancy relative to Richardson's interpretation here.

6. Page 1 Line 40: solved analytically.

7. Page 1 Line 48: what do the authors mean by "form", do they refer to the geometry or shape of eddies and plumes? If they do, then scaling analyses along different directions could still be useful in determining such geometry or shape parameters (e.g., the determination of vertical stratification or anisotropy parameters using separate scaling analyses in the horizontal and vertical directions).

8. Page 2 Line 5: It would be better to explicitly state this rationale here (at least briefly).

9. Figure 1 caption: Obukhov length $(L)$ "as defined by Eq. 3", where $u_*$ is the friction velocity (as in Sect. 2).

10. Page 2 Line 60 Page 3 Line 28: Power Laws might even indicate generalized scale invariance in which case the underlying structures need not be self-similar (they could even be self-affine).

11. Page 2 Line 68: cite these previous works and briefly explain what has been attempted in these works.

12. Page 3 Line 8: It would be better if friction velocity is briefly defined here.

13. Page 3 Line 31: mention the scales replacing the scales in Eq. 2.

14. Page 3 Line 17 Line 42: If the authors imply that $u_* = 0$ by the term windless convection, it is better to mention this explicitly.

15. Page 3 Line 78: briefly define inversion height here.

16. Page 3 Line 85: inner-scaled ($\zeta = z/z_s$) and outer-scaled ($\zeta = z/\lambda$) versions.

17. Page 4 Line 35: briefly define the mixed layer here.

18. Section 3.3: the heading should be "plume mean velocity variance", and again in Page 4 Line 86, it must be plume mean velocity variance.

19. Page 5 Line 27, 28: the authors neither define the primed variables $w', \theta'$ (how are they related to the original variables $w, \theta$) nor mention the significance of using them here. Do they indicate perturbation or fluctuation of the variables, why do they appear only in the heat flux equation?

20. Page 5 Line 57: How is the speed of the plume relevant in Eq. (23)?

21. Page 5 Line 71: wouldn't a horizontal cross-section of the 2T model show a two-dimensional comb where the third dimension (height) represents the temperature

(if the temperature in each point is plotted), and won't there be teeth of two different heights ($\theta_u$ and $\theta_d$) rather than a single height, as mentioned in Line 72. The authors could provide a schematic diagram of this transect.

22. Page 6 Line 11: It isn't quite apparent how the 2T model explains this.

23. Page 6 Line 51: further explanation will be helpful here.

24. Figure 2 caption: $\theta_p$ has to be defined properly as the temperature corresponding to the peak value of the PDF (to avoid it being confused with the peak value of $\theta$).

25. Page 6 Line 72: Does the prime indicate that the variables have been averaged over a number of runs? Once again, as mentioned in comment 19 this is not clear.

26. The heading of this section (Sect. 5.1) says $T$ and $wT$ probability distributions, but Fig. 2 shows $\theta - \theta_p$ distribution (the authors must explicitly state here if both temperature and potential temperature have the same probability distributions). Fig. 3, on the other hand, shows $w' - T'$ joint probability distributions weighted by the product $w'T'$ (not the probability distribution of $w'T'$), therefore it is not clear how it can be called the wT or heat flux probability distribution.

27. Page 7 Line 9: do the authors actually refer to "length" when they mention "size" here?

28. Page 7 Line 12: it would be apter if the authors use the term scale-invariant instead of self-similar following comment 10.

29. Page 7 Line 25: the authors could specify at least approximate wavenumber or scale values of these three ranges here while defining them.

30. Page 7 Line 41: a brief explanation as to how these scales were found in those works would be beneficial here.

31. Page 7 Line 53: plumes, where the scales $l_1$ and $l_2$ are greater than $l_3$.

32. Page 7 Line 64: this searching seems to be physical (dimensional) interpretation based trial and error rather than a rigorous mathematical procedure.

33. Page 7 Line 78: needs a mathematical explanation.

34. Page 7 Line 79: contradicts the scale mentioned in Table 1.

35. When the authors say temperature or velocity scale (e.g., Page 8 Line 11) they must be implying that the statistical average of temperature or velocity fluctuation scales as this particular term, since the power-laws discussed here are statistical ones (even though they are dimensionally correct, these laws need not hold good without averaging). The authors seem to obtain (based on observations) a term for the temperature or velocity variance scale (meaning that the statistical average of the square of the temperature or velocity fluctuation scales as this term) and take square root of this term to get the temperature or velocity scale. This means that the authors assume that square of average and average of square are equal, which is only valid for negligibly small fluctuations. This issue occurs multiple times throughout the paper (e.g. Page 5 Line 21, Page 7 Line 79, Page 8 Line 26, and Line 30).

36. Page 8 Line 18: how does the length scale as this term?

37. Page 8 Line 29: do the authors mean up-plume here?

38. Section 6.2, Figure 4, and Figure 5 on Page 10, 11: Once again the undefined primed variables are used.

39. Page 10 Line 21: it is better to give the empirical relation here.

40. Page 11 Line 28: the mixed scale and doubly mixed length scale seem to match at $\lambda = z_s$ and not at $z = z_s$ as mentioned here.

41. Figures 6 and 7: how is this scale chosen? (see comment 32)

42. Page 13 Line 6: needs more explanation.

**Minor Comments:** There are only a few minor issues

1. Page 1 Line 55: previous works.

2. Page 2 Line 55: discussion of (or about) the.

3. Page 3 Line 60: is a dimensionless length.

4. Page 5 Line 44: scales as.

---

## Referee Comment (RC2) · Anonymous Referee #2 · 23 Mar 2020

**General comments**

The paper proposes a model of rising and falling plumes of constant temperatures with vertically varying area fractions to obtain expressions of vertical velocity variance, temperature variance, heat flux and the temperature gradient within and above the SFL. The authors then try to interpret the scales proposed by them earlier in terms of the area changes of the up and down plumes with height. However, no quantitative connection seems to be worked out between the variations proposed by the model and the observed variations. Neither are the earlier proposed scales shown to be connected to the results of the model quantitatively. These issues are so prevalent in

the paper that most of the proposed interpretations of the authors' scales seem mostly conjectures with no justification for the assumptions, or proof for the interpretations or connection to the model proposed by the authors. In addition, to make matters worse, the model itself is physically suspect, as shown in Specific comments 1. Further, even though the paper is mostly free of typos and grammatical errors, it is overly verbose, especially all the sections subsequent to § 4, with many a times the same ideas being repeated over and over, to make it a drudgery to read. I detail some of these issues below. Due to these issues, I do not think the paper in its present form is suitable for publication.

**Specific comments**

1. Equation 9 is not physically correct since it is not volume that is conserved but mass. Hence the terms must be multiplied by the corresponding densities of the upward and downward regions. This would imply that all the subsequent relations that use (9), like (13) are erroneous.

2. The origin of (11) and (15) and the steps leading to (12) and (16) are not clear.

3. Line 199 : 'On average the up-plumes.....'. Is this information available from earlier research?; if so, please cite.

4. There must be far more discussion in § 3.2 about the relationship of $\sigma_{\langle w \rangle}, \sigma_{\langle w \rangle}, H$ and $d\theta/dz$ with $f_u$ and $f_d$.

5. I am puzzled by the interpretation of fig 2 in § 5.1 by the authors. Figure 2 shows the probability density function of $\theta - \theta_p$ obtained from point-wise measurements over time in SLTEST. LHS of the peak value of the pdf shows the probability of occurence of $\theta < \theta_p$, while the RHS of the peak value shows the probability of occurrence of $\theta > \theta_p$. Since $\theta_p$ is the peak temperature measured from various measurements over time at a point, a temperature less or more than $\theta_p$ does not

imply that the mass of air is falling down or rising up, as the authors assume. Masses of air with $\theta < \theta_p$ could as well be rising and vice versa at any point of time since all rising or falling masses of air do not have the same temperature in these measurements, unlike that in the case of the 2T model. So to interpret this experimental PDF based on the assumption in 2T model would be wrong, since the experiments do not satisfy the assumptions made in the model.

Even if we ignore this error, the interpretations of fig 2 based on the 2T model is again problematic. Firstly I do not see that the area of the PDF to the left of the peak is larger than the area to the right of the peak, as the authors state, since the right hand side extends over a much larger range of $\theta - \theta_p$. Neither is it obvious that the areas of the PDF to the right of the peak decrease with increase $z$ since the height of the peak also increase with increase in $z$.

More importantly, the authors make the leap of faith, in line 261, of relating the areas of PDF to the fractional areas of plumes. This connection cannot be made since the fractional areas that the authors speak are the horizontal areas occupied by rising and falling regions of plumes in a horizontal plane. The PDF obtained by point wise measurements over time only reflects the time of duration that the point has the particular temperature by being within a rising or falling plume and not the horizontal area of the rising or falling plume. Even application of Taylors frozen turbulence hypothesis will only give the vertical extent of the plume, and not the horizontal areas, since the flow is predominantly vertical.

6. Line 303: How is it obvious that the plume velocity scale is $(z\epsilon)^{1/3}$ when the $T$ variance scale is $H^2(z\epsilon)^{-2/3}$? Line 304: How does the $w$ variance scale being same identify the larger eddies shape the plume? Line 289, 305:Further, what physical model or scaling argument shows that when a larger eddy process with a length scale $l_1$ organises a smaller plume with a length scale $l_2$, the resultant process has a length scale of $\sqrt{l_1 l_2}$?

7. In line 311, the authors say that the $T$ variance scale $H^2(z\epsilon_o)^{-2/3}$ is incompatible with $(\theta_u - \theta_d)$ and then conjecture another reconciliation by proposing, without any proof, composite plumes above SFL. The 2T model's prediction (16) is then anyway in contradiction with the observation, which the authors justify by saying that, had the reality had what the 2T model assumed then we would have obtained the prediction of 2T model!! So even when the model's prediction contradicts observation, it is because the actual situation is not following the assumptions in the model, and not the deficiency of the model!! In para starting at line 318, the proposed $T$ variance scale is shown to suggest a plume aggregation process which results in having a $(z/z_s)^{-1/3}$ variation of the plume area fraction. Is there any proof for such an area variation? Does the 2T model suggest such a variation of plume areas? No answer is given.

8. Second para page 14: The authors argue that the length scale $\sqrt{\lambda z}$ of $T$ spectrum is consistent with the -1/2 power law of the mean temperature profile, if the area of plumes vary as $(z/\lambda)^{-1/2}$. The up plumes are suggested to be embedded in outer eddies so that horizontal convergence and hence the plume area scales as $(z/\lambda)^{-1/2}$, without making it clear why the horizontal convergence ( what does this term mean? ) should scale as suggested. This argument is hence based on many unverified assumptions and no proof for the proposed variation of area of plumes is given. Later, it is suggested that even when the aggregation properties change to change the spectral length scale to $\lambda$ at lower wave numbers, the same power law is said to hold; this is contradictory since the argument for mid wave numbers was based on the spectral scale and when the spectral scale changes the power law of mean temperature profile is also expected to change. All of the above is just conjecture with no proof for the assumptions made or the results proposed shown.

Same is the case with the argument for the scales of $wT$ cospectra in the next paragraph, which is again based on the argument that two process of length

scales $l_1 = \sqrt{\lambda z}$ and $l_2 z$ when interact create a process of length scale $\sqrt{l_1 l_2}$; what is the physical justification for such an assumption? Why cant the new length scale be some other dimensionally consistent power law combination, like say, $l_1^{1/3} l_{2/3}$

9. Line 409:Taylor frozen turbulence using mean wind speed is not valid when there are rising and falling plume regions. No justification to show that the hypothesis is valid in convective turbulence is given. The issues pointed out above for the above SFL interpretations hold for § 6 on interpretation of scales within the SFL.

10. § 6.4 is just a rephrasing of all the ideas described earlier in the previous sections. As pointed out earlier, these are just conjectures with no justifications for the assumptions or proofs of the results being given. Mixing length models do not work in convective turbulence, here it is assumed to be valid.

**Technical corrections**

1. Kinematic heat flux must be defined in line 45 (page 2), same is the case with potential temperature in the caption of figure 1.

---

## Author Comment (AC1) · 23 Apr 2020

April 23, 2020

**1   General Comments**

The principal criticism here is that we do not provide enough context for the reader who is not already familiar with convective atmospheric boundary layers. In particular, the reviewer is clearly not familiar with the MOST similarity model widely used by the meteorological community.  His perspective has its benefits because he asks us to supply definitions and details that might be useful for readers from other disciplines, even while they may not be needed by meteorological readers. The question is how much meteorological information to include?

We chose to assume that the most readers will be familiar with scaling in turbulent flows, and wrote the paper so that such readers would not require a detailed particular knowledge of convective atmospheric boundary layers for the overall argument of the paper to make sense.  At the reviewer's request we now define many terms more carefully in the text, but we still rely on the cited sources for more background and detailed information. We hesitate to reference standard text books because all of them are written from the perspective of the statistical fluid mechanics model of turbulence, in which the actual forms and scaling properties of eddies and plumes play no part. Our approach is to regard the CBL as a dissipative system, and to understand it through its emergent properties, including the properties of eddies and plumes. We have added a

sentence to the opening paragraph that would be sufficient to orient a reader familiar with Rayleigh-Bénard convection but not CBL flows. It may also be enough for the reader with a more general systems background.

The other general criticism is that we do not take a more mathematical approach to developing our ideas. Our basic approach is laid out in the first paragraph of the paper. We use the words 'emergent properties' to describe eddies and plumes as a signal that we are reaching for things beyond the capabilities of direct mathematical analysis. Some of our responses, below, elaborate on this.

**2  Detailed Comments**

We thank the reviewer for his many detailed comments on errors and omissions. An outside eye is very useful here. We have adjusted the text to accommodate many of his points. Here we comment on some suggestions we have not followed.

6. "solved analytically". We have not added the word "analytically". The word solve means 'to find an explanation for'. In physics that explanation often takes the form of a compact mathematical expression found by analysis of the governing equations. The word 'analytically' is, at best, redundant. Adding it would have the unfortunate implication that the NS equations can be solved by some other means. What means? Not simulations: they, like experiments, provide numbers but not explanation.

7. We leave the sentence be. Whatever we can learn by scaling analyses, it is not enough to determine the exact form of these patterns of motion and temperature, especially since these patterns are often dynamic.

8. We would like to oblige, but space does not permit. Ours is not the standard model so we would have to explain and justify all of our work since 2004. Instead we give just the information necessary to introduce the arguments in this paper.

11. How to "briefly explain" the background to our paper? We have chosen just to reference our earlier papers. The reason is that there have been no close antecedents for our work beyond what is described in those papers. There have been no other reports of mixed lengths scales for any scalar properties, and no treatment of 'plumes' as entities quite distinct from 'eddies'. Our work has, of course, drawn on a great deal of information from throughout the literatures of boundary-layer meteorology, fluid mechanics and complex systems, but collecting and interpreting these would take an unreasonable amount of space. We reserve that for another time, after we have assembled and interpreted more of the empirical evidence.

20. The speed of the plumes is not relevant in (23).

21. We refer to a transect, not a cross-section. We have added a sentence to explain the connection of length-fraction along a transect and area fraction on a cross sectional plane.

30. The search for scales, the three-range nature of the scales of spectra above the SFL, and the discovery of mixed and doubly mixed scales were all done empirically and in parallel with the development of a conceptual model of the roles of the various kinds of eddies and plumes in CBL turbulence. It would take a substantial amount of space to summarize that work here. Rather, we take the results in Table 1 as given information, traceable through the references, and we attempt to interpret it in terms of the current conceptual model. A basic account of this conceptual model is given in Laubach and McNaughton (op cit), as referenced in the introduction. However, some of the the empirical results in that paper were difficult to interpret, so we performed a new empirical analysis using better data from SLTEST, as reported here. Progress in the present paper is that it extends the conceptual model by elucidating the role of plume cross-sectional areas. The advantage in this development is shown by detecting the error in, and and so correcting the empirical scaling of $T$ spectra and $wT$ given in earlier work. In this we have performed another iteration of the relationship between the compact presentation of empirical results and the associated conceptual model.

31. The reviewer's observation is correct, but the method goes beyond dimensional analysis. It is nonetheless empirical and full of surprises, with constant iteration between the empirical results and the development of a conceptual model. How should we understand the parts of the system and their interactions? The present paper represents a step in that iteration, introducing the idea of the cross-sectional areas of plumes a key explanatory characteristic.

32. There is a rather deep philosophical question here about the role of mathematics in understanding particular dissipative systems. To us, science is about the detection and efficient representation of patterns in the world. Some patterns can be simple and others quite hard to recognize. Some patterns, even quite complicated ones, can be represented very efficiently by causative relationships and mathematical equations, and some not. Turbulent flows are an example where we can write the governing equations very compactly, but they cannot be solved and so give very limited information on the flow beyond gross symmetry properties. Turbulent flows are very interesting and educative systems to work with since we have exact definitions of material properties, energy, entropy and more but must still rely on empiricism to discover the transport properties of these flows. In our work we use mathematics extensively in data analysis, but recognising and interpreting the revealed patterns is a more informal process. Our case relies on how well and how generally our scaling procedures work; any informality in their discovery notwithstanding.

33. Table 1 gives the scale for the velocity variance. The velocity scale is as given.

34. The scales shown in Table 1 are based on their ability to collapse spectra and cospectra onto universal curves in each of three ranges. We look for, and frequently find, self-similar properties of eddy populations. Since the spectra are only as good as the quality of the samples, each spectrum and cospectrum is averaged over many runs to get an ensemble-mean spectrum at each height. The scales must have the dimensions temperature$^2$ or heat flux, respectively. The scale for the heat flux is clear enough since $H$ is one of our basis parameters. Indeed, $H$ is the only one of our

basis parameters to carry a temperature dimension. We can construct a temperature scale using $H$ divided by a scale velocity, and we find that $(z\epsilon_o)^{1/3}$ does the job well. It is also consistent with our conceptual model that this velocity scale represents the involvement of impinging outer Richardson eddies. In fact we went looking for this scale because $(z\epsilon_o)^{2/3}$ also scales the $u$ spectrum above the SFL. That is to say, we first reviewed our understanding of the system, guessed an answer and then confirmed that answer empirically. This is a bootstrap operation where almost nothing is given ab initio. Dimensionless ratios such as $(z/\lambda)$ can also be involved, and these are discovered when spectra or cospectra have similar shapes but do not collapse satisfactorily without them. Similarity methods have a long history in fluid mechanics, but in the present work the possibility of mixed scales makes dimensional analysis far less useful.

39. The KY90 relationship is not used in this paper, so we see no need to write it out. Indeed, the KY90 reference could be omitted. Its value is that a reader could more-easily identify the point we reference in Laubach and McNaughton (op cit).

41. See replies to points 30, 32 and 34.

42. The vertical velocity in an attached eddy must go to zero at the ground and at its top, and be maximum somewhere between but not too near either extreme. The $\sim$ symbol traditionally means of the same order as. The key point here is not the factor 2 but the reminder that the height scale is $z$. It seems unnecessary to make a big point of this, but equally we judge that the reminder does serve a purpose and should be left in.

---

## Author Comment (AC2) · 23 Apr 2020

April 23, 2020

**1 Overview**

We thank the reviewer for his careful reading of our MS and his helpful comments. He has identified a number of places where our assumptions have not been made explicit or where our explanations are inadequate or obscure. We have attended to these matters. Making these amendments has not changed the underlying arguments in our paper.

The reviewer has also expressed a more general reservation, and it concerns the nature of our paper. Our objective is stated in the last paragraph of the introduction. It is to extend a conceptual model that can explain the empirical scaling results reported in this and previous papers. The 2T model is important because it helps us define what we mean by 'plumes'. However, it is only a Toy model, designed to clarify concepts rather than provide a basis for computation. We provide an exact definition of a plume for an idealized flow where plume boundaries are sharp and then say "real plumes are something like that". Importantly, we find that the idea of the cross-sectional areas of plumes is likely to have some validity, if not a definite value, beyond the confines of the 2T model. The reviewer thinks we should have developed the 2T model as a computational model. This lies beyond the ambition of the present paper. We address the reviewer's comments below.

**2 General Comments**

The reviewer's first sentence is not quite correct, and the subtleties of its inaccuracy have a lot to do with our differing views of what the paper should be about. Firstly, we use a Toy model, the 2T model, to develop the concept of plumes and to justify talking about the cross-sectional areas of plumes. We then go on to use this concept to develop a conceptual model for the empirically-discovered scaling properties of temperature spectra, cospectra and profiles. The reviewer has not recognized the difference in our terminology of up-plumes and down-plumes, which might, or might not, be moving upwards or downwards locally, and his terminology of upwards-moving and downwards-moving plumes. The reviewer should also acknowledge that our model is a conceptual model intended to underpin a semi-empirical similarity model, not to become a computational model. We attempt to explain why our empirical scaling results are as they are, and we extend our purview to include the mean temperature profile. In our empirical analyses of the data from the SLTEST experiment we found that velocity and temperature spectra could all be collapsed onto universal curves in each of three spectral ranges. Further, we found that length scales needed to achieve this collapse were not just the simple length scales used in the orthodox Monin-Obukhov similarity theory, but that mixed length scales and doubly mixed length scales were also required. We have achieved a very large improvement in our ability to represent $T$ spectra and $wT$ cospectra in universal ways.The scales that allowed us to do this are new to science. Our question is how should we understand this zoo of new length scales?

In general terms, our longer-term project is to provide a replacement for the standard Monin-Obukhov similarity theory (MOST). This is a statistical model based, conceptually, on Richardson's assumptions about the local action of buoyancy and the irrelevance of flow conditions at the upper limit of the atmospheric surface layer (Richardson, L.F.: Proc. Roy. Soc, Lond. A, 97:354-373, 1920). In essence, Richardson's model

is a linearized model of boundary layer turbulence—one where the each term of the RANS energy equation can be interpreted individually and without regard to context in the flow system. MOST can be regarded as Richardson's theory expressed in a more convenient form. In sharp contrast, our similarity model accepts that the atmospheric boundary layer is a dissipative system, with all that that implies about the importance of energy flows in sustaining the turbulence ( Fig. 8) and about the cyclic nature of cause and effect in complex systems. We note that similarity models, such as MOST and ours, do not provide quantitative results themselves, but they do provide a framework by which experimental results can be represented in universal ways.

Though this is our underlying purpose, we delay any general discussion of similarity models until the end of the paper, which is to say until after we have shown that the set of scale lengths that we have discovered can be explained in terms of the cross-sectional areas of plumes, and the way these areas vary with height. This connection has no precedents in the literature of boundary-layer meteorology nor, so far as we are aware, in fluid mechanics generally.

When framing our discussion the first problem was to define plumes somewhat more closely than "patterns of scalar concentration". We wanted to use the idea of the cross-sectional areas of plumes, but without a definition of a plume boundary this is as elusive as an exact definition of a plume itself. Problems of definition are common when dealing with complex systems. (What is a species in ecology?) Our approach is to define a plume in a rigorous way for a toy model, then to say "a plume is something like that". We appear to have achieved our goal inasmuch as the reviewer understands what we mean and adopts the word without complaint. However, the reviewer also wants more than just a toy model, and for us to develop the 2T model into a computational model. Such a development lies beyond our present ability and purpose.

The reviewer points out that the 2T model is incorrect as it stands. He is strictly correct since we had omitted mention of our assumption that air density is independent of temperature. This has been remedied in the revised text. The 2T model, as a toy

model, still serves our purpose.

To the reviewer's point about repetition: the same idea is repeated several times in section 3, but the details vary each time. Our purpose is to show that the various length scales shown in Table 1 and Figs. 6 and 7 can all be interpreted in terms of plumes, and the aggregation properties of plumes, in a consistent way. The point of the repetition is to show that the basic concept of composite plumes is consistent with the observed length scale in each case. While writing the paper, and while trying to achieve this consistence (aka repetition) we were lead to question some results presented in an earlier paper by J. Laubach and K.G. McNaughton (Bound. Layer Meteorol. 133:219-252, 2009), so the present paper also reports new analyses based on the more comprehensive data from the SLTEST experiment. The revised results are given in Figs 6 and 7 of the present paper. Including these has lengthened the paper beyond what was originally planned.

**3 Specific Comments**

1. The density of air is assumed independent of temperature in the 2T model, so (9) is correct. This assumption is now stated explicitly. We note that the 2T model is a Toy model, not a computational model. Its purpose is given at the start of section 3.3. With (9) accepted (11) follows from (10), and (12) then follows since $f_u(1-f_u)$ and $f_d(1-f_d)$ are both equal to $f_u f_d$, using (8). There are no tricks. [Original equation numbers.]

2. The derivations are exactly parallel to those of (11) and (12).

3. This is implied directly by (19). The comment is intended to encourage the reader takes a step back from the formalism and to think about plumes as physical entities.

4. Only formal aspects of the 2T model are set out in section 3.2. It is enough that we can define them and use them to identify scales. Particular results are referred

to, when needs be, in later sections. Implicit in our concept of plumes is that they are entities that exist in space and time. They therefore lie outside the purview of the Monin-Obukhov similarity model, in which scheme there is great confusion as to the fundamental differences between 'eddies' (patterns of motion) and 'plumes' (patterns of scalar concentration).

5. To make our point clearer we have added an equation for the mean temperature, new (10) to section 3, and have connected our interpretation of Fig. 2 to that equation explicitly in section 5.1. We have also rewritten our argument and, in particular, made clear our use of Taylor's frozen turbulence hypothesis when making our "leap of faith" in the interpretation of Fig. 2. These additions should allay the reviewer's concerns.

Throughout the paper we are very clear that that the 2T model is a toy model—useful for illustrating concepts and generating ideas but not an exact model. In a general way, and in real CBLs, we know that air is heated by contact with the ground and that this air then moves upwards, on average, to convey heat upwards into the bulk of the CBL. We can associate this, conceptually though not in detail, with the 2T idealization of up-plumes. We also know that air is entrained through the top of the CBL, and that this air is cooler, on average, than the air found close to the ground. This entrained air moves downwards, on average, throughout the CBL and we can associate this with the 2T concept of down-plumes. The question is whether that a useful thing to do? Can it help us to understand why the shapes of temperature PDFs change with height the way they do? Alternatively, what can PDFs say about real plumes?

6. The rationale for our scaling scheme is given in paragraph 3 of the Introduction. Going back to McNaughton et al (op. cit. 2007) we identify the velocity scale as the velocity scale appropriate to impinging outer Richardson eddies. The outer dissipation rate is independent of height above the ASL and the length scale is $z$ because these eddies impinge onto the ground. A more complete statement is given in the paper by Laubach and McNaughton (op. cit., 2009). Re line 287-305: we are, in effect, stating our working hypothesis here. We are trying to explain what lies behind the empirical

observations reported here and in earlier papers. Below we test that this explanation is consistent with the observations in a range of situations.

7. The paragraph has been redrafted. Composite plumes are more than just conjecture. If we watch the smoke rise from a fire, or to subside and fumigate at ground level when the surrounding air is subsiding, we see that the smoke becomes less dense as the plume grows and spreads. We must conclude that a real smoke plume is not like a fixed-concentration up-plume of the kind defined in the 2T toy model. How can the 2T model then help us understand the real world? The problem is not that the 2T model neglects molecular diffusion since this must be unimportant at the meter or ten-meter scale of the smoke plume. The problem lies in the different identities of real plumes and up-plumes. The first step towards a reconciliation is to think about real plumes as composites of 2T-type, fixed-concentration up-plumes (i.e. undiluted smokey air) and down-plumes (clear air), with the filaments of each remaining conceptually distinct but increasingly entangled nearer the ground due to the mixing action of small eddies. Maintaining the fixed identities of smokey and clear air then gives rise to the concept of composite plumes. They provide a useful approximation of reality and they allows us to align observed scaling properties with the concept of plume areas made concrete in the 2T model.

The wider problem here seems to lie with our terminology—what we mean by up-plumes, down-plumes and composite plumes. We have added a paragraph on these at the end of section 3.3, and have made small changes elsewhere where confusion might arise.

8. The original paragraph is rather turgid and has been redrafted. We have no explanation for why the power is 1/2. We are now aware that the half power law found experimentally in the mixed scales for certain dynamical properties close to smooth walls have been derived by the mathematical technique of asymptotic matching of the profiles in the viscous sublayer below with those in the log layer above (N. Afzal, J. Mécan. théor. appliq.1:963-973, 1982). The method has since been extended to matching

passive-scalar profiles near the smooth floor of a channel flow, but not yet to matching temperature in the transition between mixed-layers and log layers in CBL flows.

9. We used Taylor's frozen turbulence hypothesis because it is conventional to do so in Boundary-Layer Meteorology, and because we had no alternative. This is not to say we are unaware of its limitations. The core of the problem is Taylor's statement that eddies are carried along by the mean flow (GI Taylor, Proc. Roy. Soc. Lond. A, 164:476-490, 1938). We have known for a very long time that plumes are not simply carried along by the mean flow: their phase velocities increase with eddy size in the ASL (DS Davison, Quart. J. R. Met. Soc.. 100:572-592, 1974). Recently Cheng et al. (Geophys. Res. Lett. 44:4287-4295, 2017) confirmed this by the splendid innovation of measuring temperatures along an optical fibre strung out horizontally over the ground. We have yet to incorporate their results into our interpretations of time series, but insofar as corrections for plume velocities can be scaled using the same set of similarity variables as we use elsewhere, the revision should entail just a change in the shapes of the spectra and cospectra, not a change in their scaling properties or universality. Our conceptual model will remain intact, though Figs. 6 & 7 will require adjustment. We endorse Cheng et al's point that a distorted wavenumber axis in our Fig. 7 will lead to an underestimation of the area under the $wT$ cospectrum, and so to significant 'flux loss'.

10. The section has been rewritten and retitled to reduce repetition. The relevance of the mixing length model is now explained much more fully. The new material is important because it emphasizes the failure of Reynolds' analogy between momentum and heat transport.

**4  Technical comments**

All terminology is standard for those familiar with transport of scalars in the atmosphere. For others we have checked that Wikipedia is a sufficient resource.